# The reductive glycine pathway allows autotrophic growth of *Desulfovibrio desulfuricans*

Irene Sánchez-Andrea [1✉], Iame Alves Guedes [1,7], Bastian Hornung[2,7], Sjef Boeren [3], Christopher E. Lawson[4], Diana Z. Sousa [1], Arren Bar-Even [5,8], Nico J. Claassens [1,5✉] & Alfons J. M. Stams [1,6✉]

Six $CO_2$ fixation pathways are known to operate in photoautotrophic and chemoautotrophic microorganisms. Here, we describe chemolithoautotrophic growth of the sulphate-reducing bacterium *Desulfovibrio desulfuricans* (strain G11) with hydrogen and sulphate as energy substrates. Genomic, transcriptomic, proteomic and metabolomic analyses reveal that *D. desulfuricans* assimilates $CO_2$ via the reductive glycine pathway, a seventh $CO_2$ fixation pathway. In this pathway, $CO_2$ is first reduced to formate, which is reduced and condensed with a second $CO_2$ to generate glycine. Glycine is further reduced in *D. desulfuricans* by glycine reductase to acetyl-P, and then to acetyl-CoA, which is condensed with another $CO_2$ to form pyruvate. Ammonia is involved in the operation of the pathway, which is reflected in the dependence of the autotrophic growth rate on the ammonia concentration. Our study demonstrates microbial autotrophic growth fully supported by this highly ATP-efficient $CO_2$ fixation pathway.

[1] Laboratory of Microbiology, Wageningen University & Research, Stippeneng 4, 6708 WE Wageningen, The Netherlands. [2] Leids Universitair Medisch Centrum (LUMC), Albinusdreef 2, 2333 ZA Leiden, The Netherlands. [3] Laboratory of Biochemistry, Wageningen University & Research, Stippeneng 4, 6708 WE Wageningen, The Netherlands. [4] Department of Civil and Environmental Engineering, University of Wisconsin—Madison, Madison, Wisconsin, USA. [5] Max Planck Institute of Molecular Plant Physiology, Am Mühlenberg 1, 14476 Potsdam-Golm, Germany. [6] Center of Biological Engineering, University of Minho, Campus de Gualtar, 4710-057 Braga, Portugal. [7] These authors contributed equally: Iame Alves Guedes, Bastian Hornung. [8] Deceased: Arren Bar-Even. ✉email: irene.sanchezandrea@wur.nl; nico.claassens@wur.nl; fons.stams@wur.nl

**D**esulfovibrio is a genus of sulphate-reducing bacteria that is ubiquitous in oligotrophic and eutrophic environments[1]. These bacteria couple the reduction of sulphate with the oxidation of a variety of electron donors such as lactate, ethanol, formate and hydrogen, but not acetate[2]. Growth of *Desulfovibrio* sp. on formate and hydrogen was thought to be dependent on the presence of an organic carbon source (e.g., acetate)[2–7]. However, *D. desulfuricans* strain F1[8] and the *Desulfovibrio* strains HRM1 and P23[9,10] were reported to grow autotrophically, though insight into the $CO_2$ fixation pathway was not obtained.

In a previous study, *D. desulfuricans* strain G11 (hereafter *D. desulfuricans*) grew syntrophically with the methanogen *Methanobrevibacter arboriphilus* strain AZ (DSM 744) on formate as carbon and energy source[11]. Based on this co-culture experiment, it could be hypothesised that *D. desulfuricans* either is able to grow autotrophically (or formatotrophically), or alternatively that it uses an organic compound excreted by the co-cultured partner. Autotrophic growth had never been demonstrated for a pure culture of *D. desulfuricans* strain G11.

Here, we were able to grow *D. desulfuricans* autotrophically, and investigated the $CO_2$ fixation pathway involved in autotrophic growth. Genomic, transcriptomic, proteomic and metabolomic analyses reveal that *D. desulfuricans* assimilates $CO_2$ via the reductive glycine (rGly) pathway, a previously proposed yet unconfirmed seventh $CO_2$ fixation pathway.

## Results

### *D. desulfuricans* is able to grow autotrophically.

We transferred *D. desulfuricans* to chemolithoautotrophic conditions in a basic anaerobic mineral medium[12] with sulphate and a gas phase consisting of 80% $H_2$ and 20% $CO_2$ (further referred to as autotrophic conditions). The inoculum was a culture growing in chemolithoheterotrophic conditions (acetate/$H_2$/$CO_2$/sulphate, further referred to as heterotrophic). After the initial transfer, growth was observed, but only after a long lag phase (>6 days) and with a low yield ($OD_{600} < 0.12$). However, after several transfers, the growth characteristics in autotrophic conditions improved and reached those observed in heterotrophic conditions with a doubling time of ~24 h (Fig. 1 and Supplementary Data 1). Periodical transfer of the autotrophic cultures in the late exponential phase (with 10% of inoculum) successfully avoided long lag phases. We investigated if laboratory evolution took place along long-term cultivation. Hence, we again transferred a cell population, which was transferred heterotrophically for 3 years and not experienced autotrophic conditions before, to autotrophic conditions. During the first transfer it showed again a long lag phase, but after a second autotrophic transfer in the late exponential phase, a short lag phase and fast growth were observed, similar to the growth of the autotrophic culture that was transferred over 3 years (Supplementary Data 1). This indicates that the long-term transfer is not needed to reach fast autotrophic growth and that this phenotype is not based on genetic mutations. Indeed, no genetic mutations could be identified in the genome sequence of the autotrophic culture after 35 transfers when compared with the heterotrophic culture (both after 3 years of subcultivation), confirming that no laboratory evolution took place. We also transferred the autotrophic culture to formatotrophic conditions (formate/$CO_2$/sulphate) and observed good growth (Supplementary Data 1). The purity of the cultures was routinely confirmed using microscopy and sequencing of 16S rRNA gene.

### No known $CO_2$ fixation route is fully encoded in the genome.

We studied the genome of *D. desulfuricans* to identify genes involved in autotrophy, including possible metabolic routes for $CO_2$ fixation. Analysis of the genome of *D. desulfuricans* confirmed the presence of most of the common biosynthesis pathways, including those for synthesis of amino acids and vitamins (Supplementary Data 2 and Supplementary Table 1). However, the bacterium lacks key enzymes of the known $CO_2$ fixation pathways[13,14], including ribulose-1,5-bisphosphate carboxylase/oxygenase (Rubisco) of the Calvin cycle and acetyl-CoA synthase/carbon monoxide dehydrogenase of the reductive acetyl-CoA pathway (Supplementary Table 2). As the *D. desulfuricans'* genome lacks malate dehydrogenase and succinyl-CoA synthetase, it cannot operate the reductive TCA cycle[15] or any of its variants[16,17]. The lack of genes for known pathways for $CO_2$ fixation is also observed in other *Desulfovibrio* members (Supplementary Data 3). Indeed, most *Desulfovibrio* members do not grow autotrophically with hydrogen, with a few exceptions mentioned above[8–10]. The observed autotrophic growth of *D. desulfuricans* indicates the presence of an unknown $CO_2$ fixation pathway in this microorganism.

### Comparative omics suggest $CO_2$ fixation via glycine.

To elucidate the identity of the $CO_2$ fixation pathway without solely relying on genome annotation, we used a comparative multi-omics approach. We first compared the transcriptome and proteome of cells grown autotrophically ($H_2$/$CO_2$/sulphate) and heterotrophically (acetate/$H_2$/$CO_2$/sulphate). We consistently found several genes and proteins that were highly upregulated under autotrophic conditions (Figs. 2, 3, Supplementary Fig. 1 and Supplementary Data 4, 5).

Highly overexpressed genes include those encoding the glycine cleavage system (GCS, DsvG11_0325-0328), the glycine reductase (GR) complex (DsvG11_1441-1448) as well as formate–tetrahydrofolate (THF) ligase (FTL, DsvG11_3068), acetyl-CoA synthetase (ACS, DsvG11_2043) and genes encoding proteins associated with ammonia limitation (GS-GOGAT system). The same genes were also upregulated in autotrophically grown cells when compared with cells grown heterotrophically with lactate as sole energy source (without $H_2$) (Supplementary Fig. 1). Hence, we propose that these upregulated genes are involved in autotrophic $CO_2$ fixation as depicted in Fig. 3 and Supplementary Fig. 2.

The GCS is a ubiquitous complex that catalyses the reversible cleavage of glycine to $CO_2$, methylene-THF and ammonia[18] ($\Delta_r G' = +4.6$ kJ per mol at pH 7.5, ionic strength 0.25 and all reactants at 1 mM[19]). The GCS is known to operate in the reductive carboxylation direction in various anaerobic microorganisms as a sink of electrons[20–22], thus producing glycine from $CO_2$ and methylene-THF. FTL catalyses the condensation of formate with THF and, together with the bifunctional methylene-THF dehydrogenase/cyclohydrolase (MTCD), can convert formate to methylene-THF, the direct substrate of GCS for reductive carboxylation. Hence, the combined reductive activity of FTL, MTDC and GCS can result in the conversion of formate, ammonia and $CO_2$ to glycine.

For $CO_2$ to be converted to glycine, it should be first reduced to formate. The molybdenum-containing formate dehydrogenase (FDH) of *D. desulfuricans* (DsvG11_0566-0569) was described to catalyse $CO_2$ reduction to formate with very good kinetic parameters ($k_{cat}$ of 46.6 s$^{-1}$, $K_m$ of 15.7 μM)[23]. FDH is highly expressed at all conditions tested (Supplementary Data 4), while some of its subunits are upregulated during autotrophic growth. Moreover, the observed accumulation of formate in the medium under autotrophic conditions indicates the reductive activity of FDH (Fig. 1a). Overall, these findings suggest that *D. desulfuricans* converts $CO_2$ to glycine via the combined operation of FDH, the THF-dependent enzymes and the enzymes of the GCS.

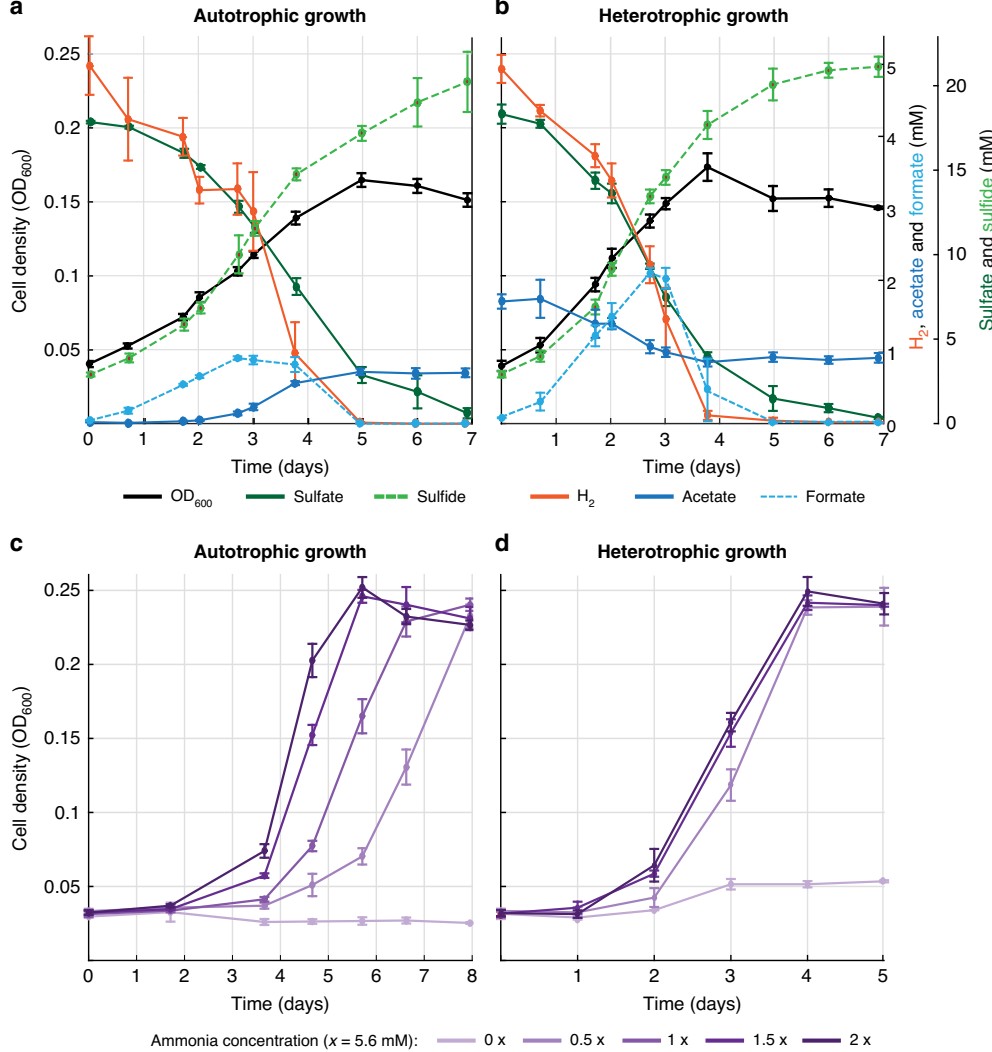

**Fig. 1 Comparison of growth of *D. desulfuricans* under autotrophic and heterotrophic conditions.** Autotrophic growth ($H_2/CO_2$/sulphate) after some adaptation transfers (**a**) is similar to heterotrophic growth (acetate/$H_2/CO_2$/sulphate (**b**). Increasing ammonium concentrations do have an effect on the autotrophic growth (**c**), but not on heterotrophic growth (**d**). All growth experiments were performed at 30 °C and 175 rpm in triplicates in 250 ml glass bottles containing 100 ml anoxic minimal medium. Error bars represent the standard deviation. Source data are provided with this paper, and raw data can be found in Supplementary Data 1.

Assimilation of glycine into the central metabolism could proceed via two main routes: (i) glycine conversion to acetyl-phosphate, catalysed by GR (DSVG11_1441-1448), which can be further converted to acetyl-CoA and pyruvate; (ii) glycine conversion to serine via serine hydroxymethyltransferase (SHMT, DsvG11_2276). Serine could be further assimilated into the central metabolism via several routes, the most direct one employing serine deaminase (SDA, DsvG11_1577) to form pyruvate. Alternatively, serine could be assimilated via conversion to phospho-glycerate, which could proceed via serine transamination to hydroxypyruvate (e.g., by pyruvate:serine transaminase DSVG11_0309 or other transaminases), subsequent reduction of hydroxypyruvate to glycerate (potentially by 2-hydroxyacid dehydrogenases DSVG11_0256 or DSVG11_0961) and finally the generation of phospho-glycerate by glycerate kinase (DSVG11_0656 or DSVG11_1884). During autotrophic growth, assimilation probably proceeds primarily via the glycine reduction route rather than the serine route, as GR was highly upregulated, while SHMT, SDA and alternative serine assimilation enzymes were not upregulated and expressed at a lower level than GR.

Acetyl-phosphate generated from glycine can be converted to acetyl-CoA either directly, via phosphoacetyltransferase (PTA, DsvG11_0941), or indirectly, via the combined activity of acetate kinase (ACK, DsvG11_0942) and acetyl-CoA synthetase (ACS, DsvG11_2043). The high up-regulation of ACS under autotrophic conditions, and the accumulation of acetate in the growth medium, indicate that the indirect route is the major pathway of acetyl-CoA biosynthesis. Finally, acetyl-CoA can be converted to pyruvate via the activity of pyruvate:ferredoxin oxidoreductase (PFOR, DsvG11_0940), which was highly expressed under all growth conditions in the transcriptome, as well as abundant in the proteome. Pyruvate can also be synthesised by condensation of acetyl-CoA and formate[24]. *D. desulfuricans* possesses two putative pyruvate-formate lyases (PFL, DSVG11_0854 and DSVG11_2562), of which DSVG11_0854 is slightly upregulated at transcript level in autotrophic conditions (Supplementary Data 4 and 5). However, as both PFL enzymes are overall having low expression levels at transcript level and not detected in the proteome, we assume they play a minor or no role in pyruvate synthesis.

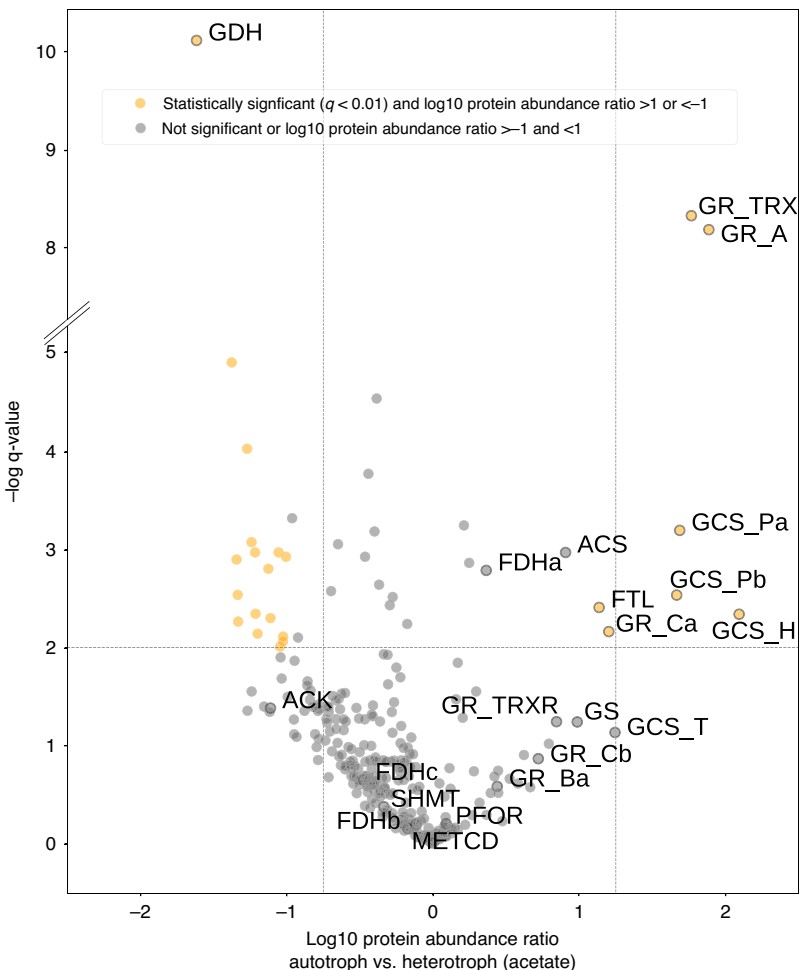

**Fig. 2 Volcano plot for protein up-regulation of autotrophic vs. heterotrophic growth conditions.** Autotrophic ($H_2/CO_2$/sulphate) and heterotrophic (acetate/$H_2$/$CO_2$/sulphate) cultivations were performed in four biological replicates to perform proteomic analysis. Growth conditions were 30 °C and 175 rpm in 250 ml glass bottles containing 100 ml anoxic minimal medium. ACK (acetate kinase); ACS (acetyl-CoA synthetase); FDHa, FDHb, and FDHc (formate dehydrogenase: alpha, beta and gamma subunits); FTL (formate–tetrahydrofolate ligase); METCD (methenyltetrahydrofolate cyclohydrolase); GCS_Pa and GCS_Pb (glycine dehydrogenase/decarboxylating, subunit 1 and 2, glycine cleavage system P-protein); GCS_H (glycine cleavage system H-protein); GCS_T (aminomethyltransferase, glycine cleavage system T-protein); GDH (glutamate dehydrogenase); GR_A (glycine reductase complex, component A); GR_Ba (glycine reductase complex; component B, subunit beta); GR_Ca and GR_Cb (glycine reductase complex, component C, subunit alpha and beta); GR_TRX (thioredoxin domain); GR_TRXR (thioredoxin-disulphide reductase); GS (glutamate-ammonia ligase or glutamine synthetase); METCD (methenyltetrahydrofolate cyclohydrolase); PFOR (pyruvate: ferredoxin oxidoreductase) and SHMT (serine hydroxymethyltransferase). Source data are provided with this paper, and complete proteomics data can be found in Supplementary Data 5.

The observation that besides formate about 1 mM of acetate is excreted during autotrophic growth is remarkable (Fig. 1), as the addition of 2 mM acetate results in heterotrophic growth. Currently, it is not clear at which concentration the switch from heterotrophic to autotrophic growth and vice versa occurs. Acetate excretion was observed previously also in an autotrophic co-culture of *D. desulfuricans* strain F1 with the non-autotrophic dechlorinating bacterium *Dehalococcoides mccartyi*. The $CO_2$ assimilation pathway in *Desulfovibrio* strain F1 was not investigated, but it produced formate and acetate when grown autotrophically[8], suggesting that it may use the same pathway as described here for *D. desulfuricans* strain G11. Further research is needed to shed light on the metabolic and ecologic significance of acetate secretion.

**Autotrophic growth improves at higher ammonia concentration.** Remarkably, genes related to ammonia limitation[25] were also strongly upregulated when grown autotrophically, although the same ammonia concentration was provided in both heterotrophic and autotrophic growth conditions. These include an ammonia transporter (AT, DsvG11_2780), the nitrogen-regulatory protein (DsvG11_2781), glutamine synthetase (GS, DsvG11_2782) and glutamine-oxoglutarate aminotransferase (GOGAT, DsvG11_2965-67) (Supplementary Fig. 2 and Supplementary Data 4, 5). The up-regulation of the AT may be related to the requirement of the GCS for a high intracellular concentration of ammonia: the kinetically and thermodynamically less favourable, reductive, glycine-producing direction of the GCS could benefit from higher intracellular ammonia concentrations[18]. However, it is not completely clear why the GS-GOGAT ammonia assimilation route would be upregulated and the glutamate dehydrogenase (GDH) down-regulated under autotrophic conditions. The ammonia concentrations used in both autotrophic and heterotrophic experiments are likely sufficient to operate ammonia assimilation via the more energy-efficient GDH route. Possibly, GS and GOGAT are upregulated together with the AT in autotrophic conditions as they are in the same operon.

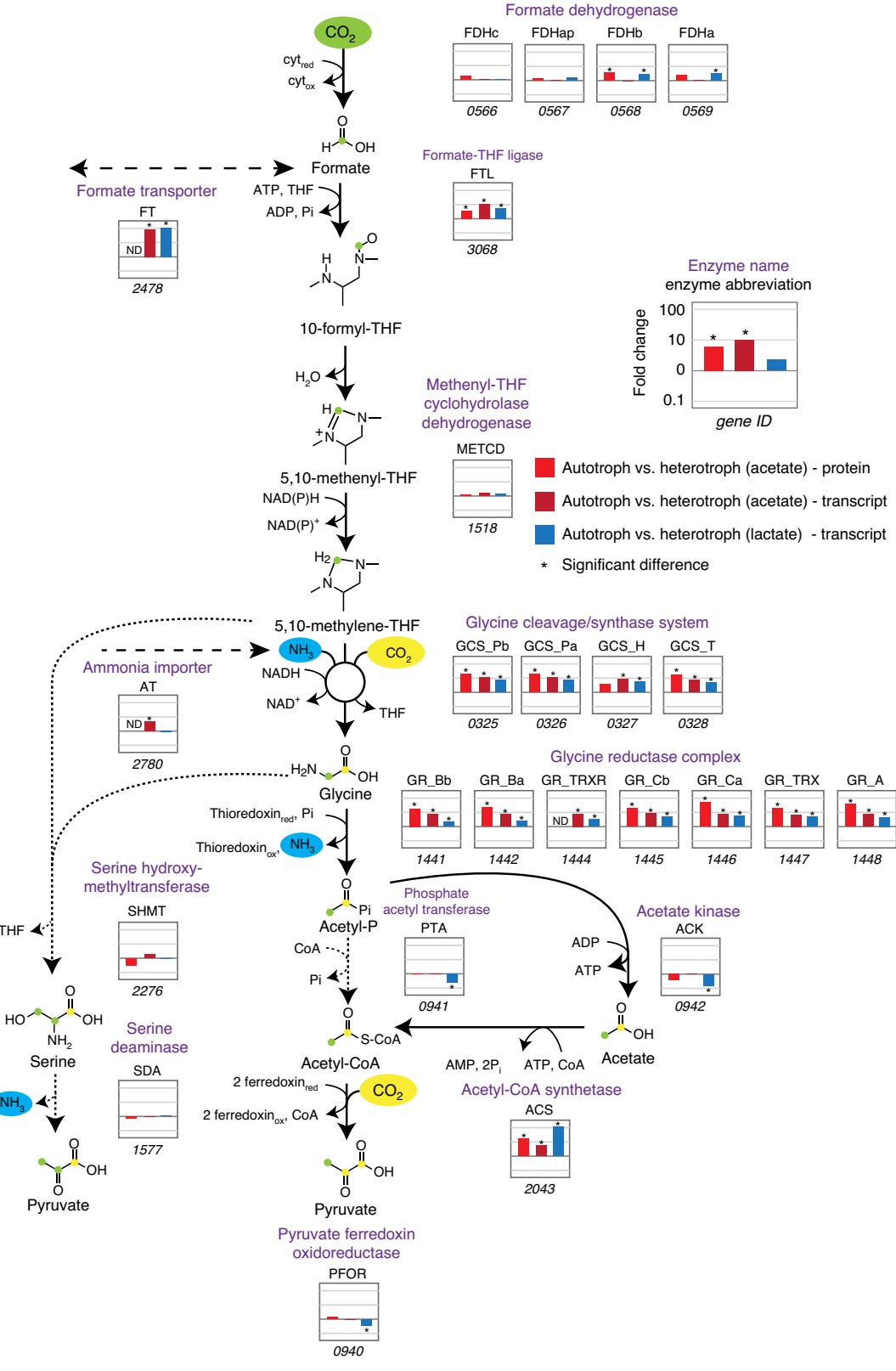

Based on these observations, we hypothesised that autotrophic growth of *D. desulfuricans* is strongly dependent on the concentration of ammonia provided in the medium. We tested the effect of ammonia on growth and found that, while heterotrophic growth was not affected by ammonia concentrations (Fig. 1d and Supplementary Data 1), the growth rate under autotrophic conditions increased with increasing ammonia (Fig. 1c). Importantly, the final biomass yield under autotrophic conditions did not change with increasing ammonia concentrations. This is to be expected as excess ammonia was not assimilated to biomass, but ammonia is temporarily fixed by the GCS and then released by GR.

**Fig. 3 Comparative omics for genes and proteins involved in the reductive glycine pathway.** Plots per enzyme represent the log10-fold change in autotrophic condition ($H_2/CO_2$/sulphate) versus heterotrophic condition (acetate/$H_2/CO_2$/sulphate) and versus heterotrophic growth on lactate as sole energy source (lactate/$CO_2$/sulphate), for both proteome and transcriptome analysis. Long-dashed lines are the transporters involved, short dashes incidate the alternative variant route via serine. Cultures were performed in four biological replicates. Growth conditions were 30 °C and 175 rpm in 250 ml glass bottles containing 100 ml anoxic minimal medium. Abbreviations are described in the legend of Fig. 2, with the addition of: ATP/ADP (adenosine tri/ diphosphate); CoA (co-enzyme A); $cyt_{red/ox}$ (reduced/oxidised cytochrome); FDHap (formate dehydrogenase, accessory protein); FT (formate transporter); GR_Bb (glycine reductase complex; component B, subunit alpha); NAD(P)H (nicotinamide adenine dinucleotide (phosphate)); SDA (serine dehydratase-like); THF: tetrahydrofolate; ND: not detected. Source data are provided with this paper, and complete transcriptomics and proteomics data can be found in Supplementary Data 4 and 5, respectively.

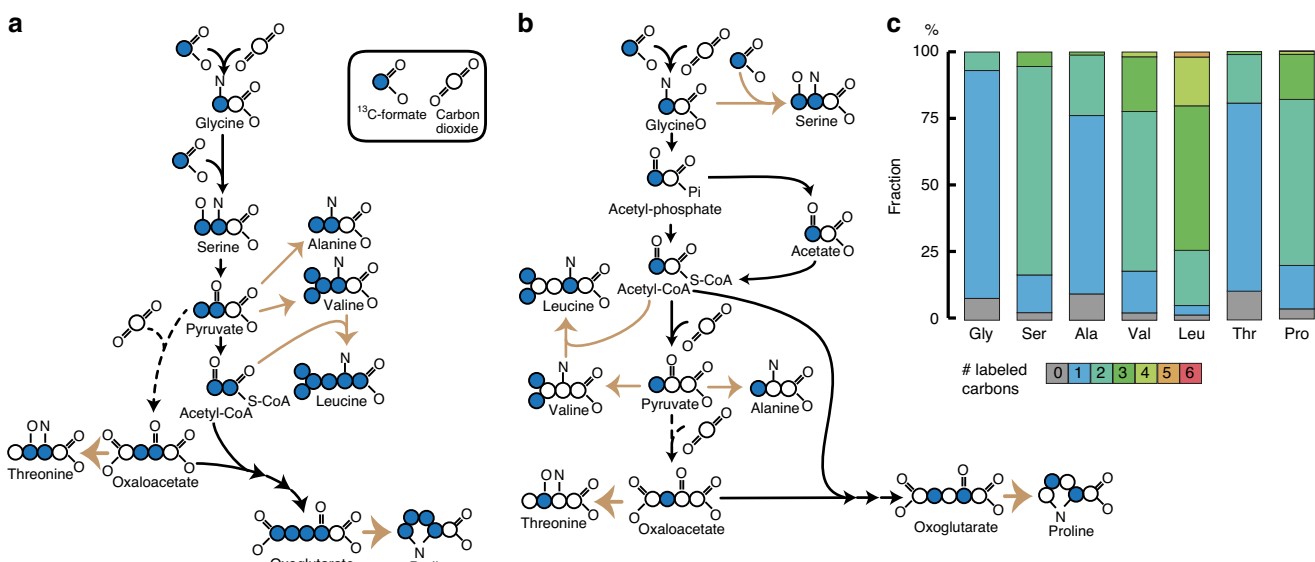

**Fig. 4 Confirmation of the reductive glycine pathway via $^{13}C$-labelled metabolomics.** $^{13}C$-labelled carbon signature expected in the serine route (**a**), glycine route (**b**) and obtained pattern of *D. desulfuricans* grown in autotrophic conditions (**c**) with addition of $^{13}C$-formate. Autotrophic cultures ($H_2/CO_2$/ sulphate) were supplemented with 75 mM sodium $^{13}C$-formate (99% $^{13}C$, Sigma-Aldrich) and grown in 30 mL bottles in 10 mL anoxic medium, cultivations were performed in biological triplicates. The dotted arrows indicate anaplerotic production of oxaloacetate via pyruvate carboxylase. Gly (glycine); Ser (serine); Ala (alanine); Val (valine); Leu (leucine); Thr (threonine); Pro (proline). Source data are provided with this paper, and data can be found in Supplementary Data 6.

**Metabolomics confirms the proposed rGly pathway.** All the enzymatic steps in the proposed pathway from $CO_2$ to glycine are reversible, and were previously proven in vitro to catalyse the reactions in the direction required for $CO_2$ fixation[26,27]. To demonstrate the operation of proposed $CO_2$ fixation pathway in vivo, we performed $^{13}C$-labelling experiments. We cultivated *D. desulfuricans* on $H_2$, sulphate and unlabelled $CO_2$, and added the $^{13}C$-isotope form of the pathway intermediate formate. This strategy was intended to generate partially labelled proteinogenic amino acids. As some of the amino acid carbons originate from formate while others from $CO_2$, the labelling pattern of different metabolites depends on their metabolic origin (if labelled $CO_2$ would be used instead, all metabolites would be fully labelled, thus providing no information on their metabolic origin). A key signature of this pathway would be the generation of single-labelled glycine and double-labelled serine, as indeed was observed (Fig. 3). The labelling of pyruvate and acetyl-CoA is expected to differ according to the assimilation route of glycine: the serine route is expected to produce double-labelled acetyl-CoA and pyruvate (Fig. 4a), while the GR route should generate single-labelled acetyl-CoA and pyruvate (Fig. 4b). The labelling pattern of alanine, valine and leucine—all derived from pyruvate and acetyl-CoA—will accordingly reflect the pathway by which glycine is further converted (Fig. 4a, b). The labelling pattern that

was observed (Fig. 4c) confirms that the GR route (together with PFOR) is the primary one for carbon assimilation in our strain. Furthermore, we analysed the labelling pattern of threonine and proline, which reflect the labelling of their precursors oxaloacetate and oxoglutarate, respectively. The labelling pattern of these two amino acids also supports the activity of GR (Fig. 4a, b). The labelling of proline and threonine also confirms the absence of a functional (reductive) TCA cycle, which would have shuffled the labelling of oxaloacetate and oxoglutarate. In fact, the labelling pattern of the amino acids shows that $CO_2$ is not fixed via the reductive TCA cycle, in which case we would expect to observe only partial labelling of all amino acids, as formate is not directly assimilated and labelling could originate only from the assimilation of $CO_2$ derived from formate oxidation.

Tracer experiments were also performed by adding 15 mM of $^{13}C$-formate to cells at the mid-exponential phase of autotrophic growth, and determining dynamic label incorporation in metabolites. Consistent with formate incorporation into *D. desulfuricans*' metabolome, intracellular formate pools were ~90% $^{13}C$-enriched after 0.5 h of $^{13}C$-label introduction (Supplementary Fig. 3A and Supplementary Data 6). In agreement with the proposed pathway, the concentrations of single-labelled glycine (Supplementary Fig. 3B) and of double-labelled serine (Supplementary Fig. 3C) increased over time. Only single-labelled

isotopomers of phosphoenolpyruvate, aspartate and 3-phospho-glycerate (Supplementary Fig. 4E–G) were observed to increase during the experiment, confirming that pyruvate was not formed by serine deamination, but rather via GR. This is consistent with the single-labelled intracellular acetate pool (Supplementary Fig. 3D) also increasing over time, as well as the low level of SDA in the transcriptomics and proteomics data and the high abundance of the GR. Remarkably, glycine and acetate, two metabolites expected to label first based on the proposed pathway (Fig. 3), labelled slower than several central carbon metabolites (Supplementary Fig. 3). One potential explanation could be substrate channelling through the pathway enzymes, where pathway intermediates such as glycine might be immediately passed to downstream enzymes without equilibrating with the bulk cytoplasmic solvent, as recently reviewed[28]. Direct channelling of glycine, keeping its concentrations low, would be thermodynamically beneficial for the operation of the pathway.

## Discussion

The pathway used for $CO_2$ fixation by *D. desulfuricans* corresponds to the rGly pathway, which was shown before to serve as a sink for reducing power, but not for autotrophic growth[20–22,29–31]. The rGly pathway was proposed to support autotrophic growth[32], but its operation for this purpose was never confirmed. Recently, it was suggested that the anaerobic phosphite oxidiser "Candidatus Phosphitivorax anaerolimi" uses the rGly pathway for $CO_2$ fixation in an enrichment culture[33]. However, this was only inferred from metagenomic data, i.e., the absence of genes from any known $CO_2$ fixation pathway and the presence of almost all components of the rGly pathway besides GR (which could be replaced by SDA). Unfortunately, in that study autotrophic growth was not demonstrated for the enrichment culture as a medium with organic carbon compounds (cysteine and rumen fluid) was used and studies with labelled $CO_2$ were not done, neither were any functional omics experiments performed. As "Candidatus Phosphitivorax anaerolimi" has yet to be isolated and no physiological evidence was provided for the activity of the suggested route, it remains unclear if this bacterium indeed grows autotrophically and uses the rGly pathway. Very recently, it was observed that the acetogen *Clostridium drakei* combines the GR variant of the rGly pathway with the reductive acetyl-CoA pathway during autotrophic growth on $H_2/CO_2$ in the presence of yeast extract[34]. As the reductive acetyl-CoA pathway is not complete in *D. desulfuricans*, the rGly pathway is the sole carbon fixation pathway in this bacterium. In summary, our and other recent findings suggest that the rGly pathway is phylogenetically widespread allowing autotrophic growth in Gram-negative bacteria as shown here for *D. desulfuricans*, and in Gram-positive bacteria, such as anaerobic clostridia (in cooperation with the reductive acetyl-CoA pathway)[33,34].

Most of the enzymatic components of the rGly pathway are present in many microorganisms[29]. While GR is quite a unique enzyme, it could be replaced by the serine route. As oxygen-tolerant variants of FDH are known to catalyse $CO_2$ reduction[35,36], the rGly pathway could also be active in aerobic microorganisms. To investigate if the enzymes of the different rGly variants are naturally present in other microorganisms, a tblastx search was performed for all relevant genes and operons in all genomes in the NCBI NT database. We found that the complete set of genes of the rGly pathway for the serine route was present in genomes of 203 microorganisms, and for the route via GR in 30 (Supplementary Data 7). Genes to support both routes are present simultaneously in 21 genomes, which included a few

members of the *Desulfovibrio* genus but also *Desulfomonile*, *Desulfosporosinus* and some *Clostridium* strains, including the aforementioned *C. drakei*. Given the diverse types of FDHs in nature, some may be missed by our tblastx analyses; when FDH is excluded from the analyses, up to 557 and 174 genomes harbour all other genes of the serine and glycine variants of the rGly pathway, respectively (Supplementary Data 7) with 157 harbouring both. Many microorganisms listed are known as heterotrophs, or acetogens assumed to use the reductive acetyl-CoA pathway. Our genome analysis suggests that these microorganisms can potentially utilise the rGly pathway for autotrophic growth, but this cannot be concluded only based on genome annotations, and requires further study.

The rGly pathway is one of the most energy-efficient routes for $CO_2$ fixation, as it consumes only 1-2 ATP molecules for the biosynthesis of pyruvate, similarly to the reductive acetyl-CoA pathway that consumes 1 ATP per pyruvate (we note that the latter pathway can additionally generate ATP via bifurcation mechanisms, see also Supplementary Table 3). Acetate can be synthesised without net ATP investment when the rGly pathway proceeds via GR and PTA, and this variant can even generate net ATP if it operates in concert with bifurcating enzymes, as observed for $CO_2$ fixation via the reductive acetyl-CoA pathway in acetogenic bacteria[37].

Due to this efficiency and to the thermodynamic feasibility of all its steps, the rGly pathway was suggested as a synthetic route for $CO_2$ fixation and formate assimilation, also in aerobic conditions[32,38]. Successful modular engineering of the rGly pathway via serine has been demonstrated recently in several biotechnological hosts: *Escherichia coli*, *Cupriavidus necator* and yeast[39–42]. This shows the feasibility of the operation of this pathway and its potential for biotechnological applications.

To conclude, we demonstrate here that *D. desulfuricans* can grow autotrophically and formatotrophically, as previously hypothesised[11]. $CO_2$ fixation is performed via the rGly pathway, which was not yet proven to operate in nature to support full autotrophic growth. Further meta-omics and physiological studies are required to elucidate the role of this $CO_2$ fixation pathway and its different variants in the microbial world and its impact on the global biogeochemical carbon cycle.

## Methods

**Strain used and cultivation conditions.** *Desulfovibrio desulfuricans* strain G11 (DSM 7057) was retrieved from our own culture collection at the Laboratory of Microbiology (Wageningen University & Research). Unless otherwise indicated, it was grown in 250 ml glass bottles containing 100 ml anoxic medium. The medium was composed of basal bicarbonate-buffered medium containing the following components (in gram per liter) Na$_2$HPO$_4$·2H$_2$O, 0.53; KH$_2$PO$_4$, 0.41; NH$_4$Cl, 0.3; CaCl$_2$·2H$_2$O, 0.11; MgCl$_2$·6H$_2$O, 0.10; NaCl, 0.3; NaHCO$_3$, 4.0 and Na$_2$S·9H$_2$O, 0.48. Furthermore, acid and alkaline trace element solutions (both 1 ml per liter)] and a vitamins solution (0.2 ml per liter) were added. The acid trace element solution contained (in mM): FeCl$_2$, 7.5; H$_3$BO$_4$, 1; ZnCl$_2$, 0.5; CuCl$_2$, 0.1; MnCl$_2$, 0.5; CoCl$_2$, 0.5; NiCl$_2$, 0.1 and HCl, 50. The alkaline trace element solution contained (in mM): Na$_2$SeO$_3$, 0.1; Na$_2$WO$_4$, 0.1; Na$_2$MoO$_4$, 0.1; and NaOH, 10. The vitamin solution contained (gram per liter): biotin, 0.02; niacin, 0.2; pyridoxine, 0.5; riboflavin, 0.1; thiamine, 0.2; cyanocobalamin, 0.1; p-aminobenzoic acid, 0.1 and pantothenic acid, 0.1[12]. The basal medium was supplemented with 20 mM sulphate, and the bottles' headspace filled with H$_2$/CO$_2$ (1.5 atm, 80:20 v/v) to provide H$_2$ as electron donor and $CO_2$ as carbon source for autotrophic growth. For heterotrophic (acetate/H$_2$/CO$_2$/sulphate) conditions, cultures were additionally provided with 2 mM of acetate. Heterotrophic cultures on lactate were supplemented with 20 mM of lactate as sole energy source with N$_2$/CO$_2$ (1.5 atm, 80:20, v/v) as gas phase. Formatotrophic cultures were supplemented with 20 mM of formate as carbon and energy source with N$_2$/CO$_2$ (1.5 atm, 80:20, v/v) as gas phase. Cultures were incubated at 30 °C in the dark.

**Physiological studies and analytical methods.** Growth under autotrophic and heterotrophic conditions was monitored in cultures grown as mentioned above and shaking at 175 rpm for optimal gas transfer. An inoculum of 10% was used (at 70%

of exponential phase) from cultures grown in the corresponding autotrophic and heterotrophic conditions. Gas and liquid samples were taken to determine optical density at 600 nm ($OD_{600}$), and concentrations of sulphide, sulphate, organic acids and $H_2$. For sulphate and organic acids determination, aliquots of 1 ml of liquid sample were centrifuged for 5 min at 10000 g. Sulphate concentrations were determined using an ion chromatograph Dionex ICS 2100 (Thermo Scientific, Sunnyvale, CA) equipped with Dionex™ IonPac™ AS16 column at 30 °C and a conductivity detector. Thirty microlitre of culture liquid was combined with 970 µL internal standard (0.25 mM sodium iodide) and of this 10 µL was injected. Potassium hydroxide (22%) was used as eluent in a multistep gradient from 1 to 45 mM with a flow rate of 0.4 ml per min. A standard curve was generated in the same way as for sample preparation; the lowest concentration included was 0.5 mM sulphate, which could still be detected. Organic acids were quantified by high pressure liquid chromatography using a Shimadzu LC-2030C equipped with a Metacarb 67H column (Agilent Technologies, Santa Clara, CA), operated at 45 °C, with 0.01 N $H_2SO_4$ as eluent at a flow rate of 0.9 ml per min. Four hundred microlitre of culture liquid was combined with 600 µL of internal standard (30 mM arabinose, 0.1 M $H_2SO_4$) and 30 µL was injected. The detection of the organic acids was performed using RI detector. A standard curve was generated in the same way as for sample preparation; the lowest concentration included was 0.05 mM for both formate and acetate, which could still be detected. For the sulphide measurements, 10–50 µL of liquid samples were added to 5 ml of water with 4 mM $ZnCl_2$ and immediately measured using the methylene blue colorimetric assay[43]. Hydrogen was monitored by gas chromatography using a Compact GC 4.0 (Global Analyser Solutions, Breda, The Netherlands) equipped with a Carboxen 1010 pre-column and a Molsieve 5A column operated at 90 °C. A pulsed discharge ionisation detector operated at 110 °C. Helium was used as a gas carrier and 5% krypton was used as an internal standard. Growth was monitored by optical density at 600 nm ($OD_{600}$) using the spectrophotometer Shimadzu UV-1800.

Cultures were routinely checked for purity. For that, 1 ml of fresh culture was centrifuged, the pellet washed twice with sterile TE buffer. The pellet was then resuspended in 100 µl of TE and 1 µl used as a template for PCR. PCR was performed in reactions of 50 µl containing Green GoTaq Reaction buffer, 0.2 mM dNTPs, 0.02 U µL$^{-1}$ GoTaq DNA polymerase (Promega, Madison, USA) and 0.2 µM of forward primer 27 f (5′-AGAGTTTGATCMTGGCTCAG-3′) and 1492r (5′-TACGGYTACCTTGTTACGACTT-3′) targeting the 16S rRNA gene (M = A or C, Y = C or T). A negative control PCR without template DNA was used. The amplification programme consisted of an initial denaturation step at 95 °C for 15 min, 30 cycles of denaturation at 95 °C for 1 min, 40 s at 52 °C for annealing and elongation at 72 °C for 1 min, followed by a final extension step at 72 °C for 7 min. PCR products were purified with the Zymo DNA Clean & Concentrator kit (Zymo Research, Irvine, CA) and sequenced by Sanger sequencing at GATC Biotech (Konstanz, Germany). Sequences obtained were verified against the NCBI database via BLAST[44].

The effect of different ammonium concentrations on growth was tested in autotrophic and heterotrophic conditions (acetate/$H_2$/$CO_2$/sulphate) as mentioned above while shaking at 175 rpm. Five different ammonium concentrations were tested: 0×, 0.5×, 1×, 1.5× and 2× over the original concentration (5.6 mM). Liquid samples were taken daily to determine growth via $OD_{600}$ and free ammonium concentrations, using the Spectroquant Ammonium Test (Merck KGaA, Darmstadt, Germany) according to manufacturer's instructions. To avoid sulphide interference in the ammonium determination, equimolar concentration of $ZnCl_2$ was added followed by centrifugation for 5 min at 10000 g, the supernatant was then used for the quantification of ammonium. A standard curve from 0 to 3 mg l$^{-1}$ of ammonium was used for the calculations and samples were diluted from 10 to 100 times to fit in the detection limits of the kit.

**Reference genome analysis**. For DNA extraction, 50 ml of cell culture grown in the aforementioned medium in heterotrophic conditions (acetate/$H_2$/$CO_2$/sulphate) was collected by centrifugation for 5 min at 15000 g. Cell pellet was washed with phosphate-buffered saline (PBS). Total genomic DNA was extracted with the MasterPure™ Gram-Positive DNA Purification Kit (Epicentre, Madison, WI) according to manufacturer's instructions. Quality and quantity of the DNA were checked on agarose gels and with the Qubit® RNA Assay Kit in Qubit® 2.0 Fluorometer (Life Technologies, Carlsbad, CA). DNA was sequenced at GATC Biotech (Konstanz, Germany) with PacBio RS2. This resulted in 150292 raw reads. The genome was assembled with the smartanalysis pipeline version 2.1.1/SmrtPipe workflow version v1.85.133289, with default parameters (see Supplementary Data 8 (Params2.xml)). Overlapping ends of the resulting assembly were detected with Blastn version 2.2.31 (standard parameters, except e value of 0.0001)[44], and manually removed (8870 bp). Illumina reads were used to correct the assembly. In total 842,997 paired-end reads with a read length of 250 bp were sequenced at GATC Biotech (Konstanz, Germany) on an Illumina MiSeq. Illumina Trueseq adapters were trimmed from this data with Cutadapt v1.2.1[45] using default settings. Quality trimming was performed afterwards with PRINSEQ Lite v0.20.0[46] with a minimum sequence length of 50 bp and a minimum quality of 30 on both ends of the read and as mean quality. The resulting reads were mapped to the PacBio assembly with bowtie2, version 2.2.9[47] with default parameters, except a maximum insert size of 1500 bp. Furthermore non-mapping reads were separated. The resulting SAM file was further converted with Samtools, version 1.3.1[48], and the

assembly was corrected with Pilon version 1.22[49], with the options – –changes – – fix bases. Non-mapping reads were assembled with IDBA_UD[50] and standard parameters. All contigs, which exceeded a content of 80% of two bases combined (e.g., >80% A and T), were discarded from this assembly. All other contigs were compared with blastn (default settings, except e value of 0.0001) against the assembly, and were discarded if they had at least one matching sequence of minimum 50% of the contigs length. All remaining contigs were manually checked for their integration in the assembly.

Genome annotation was carried out with an in-house pipeline[51]. Prodigal v2.6.3 was used for prediction of protein coding DNA sequences[52], InterProScan 5.25–64 for protein annotation[53], Aragorn 1.2.38 for prediction of tRNAs and tmRNA[54] and RNAmmer v1.2 for the prediction of rRNAs[55]. CRISPRs were annotated using the CRISPR Recognition Tool v1.2[56]. EC numbers were predicted via PRIAM version March, 2015[57], and further EC numbers were derived via the GO terms of the InterproScan result. Carbohydrate active enzymes[58] were predicted with dbCAN version 5.0[59]. The genome was checked with CheckM[60] for possible missing single-copy marker genes (defined as being present only once in 97% of the originally investigated genomes). Of the 284 single-copy marker genes, 282 were detected in one instance. CheckM failed to identify the Leucyl-tRNA synthetase via PFAM domain PF13603, despite it being annotated in gene DSVG11_2270. CheckM identified two instances of glutamine-tRNA_ligase (DSVG11_0970 and DSVG11_3047). Both proteins share 76% identity, and the RNAseq indicates expression during growth on autotrophic and heterotrophic conditions, making it unlikely that one of the genes is an artefact. Duplications of tRNA-ligases/synthetases have been described before[61]. To further determine missing genes in the genome, the genome was manually curated and investigated with Pathway Tools[62]. All pathways for amino acid, vitamin and nucleic acid biosynthesis were curated, and missing genes were noted (see Supplementary Table 1 and Supplementary Data 2). Most genes or possible candidates for nearly all biosynthetic functions could be identified, with only ten having no candidate gene at all. The raw reads were uploaded to the European Nucleotide Archive (ENA) under accession number ERR2111683 for the Illumina data, ERR2111684 for the PacBio data [https://www.ebi.ac.uk/ena/browser/view/PRJEB22313] and SAMEA104278631 for the genome assembly.

**Mutation analysis of transferred cultures**. For the single nucleotide polymorphism (SNP) genotyping, we re-sequenced with Illumina NovaSeq600 the genome of both a single autotrophic and a heterotrophic culture (acetate/$H_2$/$CO_2$/ sulphate) population after 3 years of transfers. The raw reads were uploaded to the ENA under accession numbers ERR4291999 to ERR4292000 (BioProject PRJEB22313). The re-sequenced genomes were mapped to the original genome with bowtie2 v.2.3.4.1[47], and further converted to sorted BAM files with samtools v1.6[48]. Duplicates were marked with picard tools v.1.124 (http://broadinstitute.github.io/picard/) and SNPs were called with HaploTypeCaller from the GATK package, v4.1.2.0[63], with – – sample-ploidy set to 1. All vcf files were merged with bcftools v1.4[64]. The combined vcf file was filtered to only retain SNPs, which had read counts >15 (average read coverage in genome was ~150–180).

**Exploration of the presence of rGly pathway in other genomes**. Identifications of the relevant genes of the rGly pathway in other organisms were performed with tblastx against the NCBI NT database[65] (downloaded on 29.04.2020), with an e value of 0.0001 and the option —max_target_seqs 1000000. For the operons of GR, GCS and FDH, the whole operon was used for the tblastx search, and presence of an operon was evaluated with a custom script by inspecting the co-occurrence of the genes within a range of 20,000 bp to each other. For all other genes, the coding sequence of the gene was used for the tblastx search. A gene was assumed to be present if homology of 25% between the translation products of the nucleotide sequences was identified.

To check for the presence of this pathway in combination with other FDH genes, the FDH gene clusters were identified in the following way. For all other FDHs (EC numbers 1.17.1.9, 1.17.1.10, 1.17.1.11, 1.17.5.3) the KEGG entries[66] were used as starting point to identify the organisms, which harbour the first investigated instance of these genes. If genes were listed in the KEGG entry, then these were also used for the identification. For EC 1.17.1.9, the genome of *Methylosinus trichosporium* OB3b (NZ_ADVE02000001) was used. The Refseq annotation contained the gene METTRDRAFT_RS0208115 with EC number 1.17.1.9, and surrounding genes indicated the correct identification. The genes METTRDRAFT_RS0208105–METTRDRAFT_RS0208125, including intergenic regions (genomic location 1678535-1684753) were used for a tblastx search. For EC 1.17.1.10, the genome of *Moorella thermoacetica* ATCC 39073 (CP000232) was used. The genes moth_2312-moth 2314[67], including intergenic regions (genomic location 2432486-2437304) were used for a tblastx search. For EC 1.17.1.11, the genome of *Gottschalkia acidurici* strain 9a (CP003326) was used. The genes Curi_c29380-Curi_c29410 were mentioned in KEGG, with the related publication[68], but this was extended to Curi_c29330-Curi_c29410 (genomic location 3074247-3085751). For EC 1.17.5.3, the genome of *Escherichia coli* str. K-12 substr. MG1655 (NC_000913) was used, with the genes b1474-b1476 (genomic location 1547401-1551991).

The detection of the whole pathway was based on the presence of a minimum amount of genes. The pathway was considered to be present if any of the four FDHs

was present, together with the GCS, the DsvG11_3068 formate-THF ligase, the DsvG11_1518 methenyl-THF cyclohydrolase and either each of the two optional routes: (A) GR and at least one of phosphate acetyltransferase (DsvG11_0941) or acetate kinase (DsvG11_0942), or (B) glycine hydroxymethyltransferase (DsvG11_2276) and SDA (DsvG11_1577). The GCS was assumed to be present if three out of four genes could be detected, for the GR five out of seven genes (under the assumption that the two thioredoxin genes, or any other two genes, could be present at another genomic location), and for the FDH EC 1.17.2.3 four out of six genes had to be detected (under the assumption that the two genes for the molybdenum cofactor, or any other two genes, could be present at a different genomic location). The FDH EC 1.17.1.9 was assumed to be present if four out of five genes could be detected (since either accessory gene or delta subunit seemed to be regularly missing). The FDH EC 1.17.1.10 was not evaluated, since only two genes were identified in *D. desulfuricans*, which seemed to have homology with the other FDHs. The FDH EC 1.17.1.11 was assumed to be present if five out of eight genes could be detected (under the assumption that the accessory protein and the two proteins for the production of the molybdenum cofactor, or any other three genes, could be present at a different genomic location). The FDH EC 1.17.5.3 was assumed to be present if all three genes could be detected. The relevant data can be found under https://doi.org/10.6084/m9.figshare.8970689.

**Transcriptomic analysis.** For transcriptome sequencing, cells were grown under autotrophic, and two heterotrophic conditions (acetate/$H_2$/$CO_2$/sulphate and lactate/$CO_2$/sulphate) with aforementioned media. Each condition was performed in four biological replicates. For the total RNA extraction, 50 ml of culture were harvested around the middle of exponential growth phase: 62, 58 and 66% of the maximum $OD_{600}$ for autotrophic, heterotrophic (acetate/$H_2$/$CO_2$/sulphate) and heterotrophic (lactate/$CO_2$/sulphate) conditions, respectively. Cell cultures were centrifuged for 10 min at 10,000 $g$ at 4 °C and the cell pellet was washed once with 500 μl TE buffer pH 7.4. After a second centrifugation, the pellet was snap-frozen with liquid nitrogen. Bacterial cell lysis and protein precipitation were performed immediately after, using a modified protocol from the MasterPure™ Gram-Positive DNA Purification Kit (Epicentre) as follows: lysozyme incubation was performed at 37 °C for 20 min followed by the addition of 4 μl of β-mercaptoethanol (48.7%), lysis buffer and proteinase K incubation at 60 °C for 15 min. Protein precipitation was performed according to the manufacturer's instructions. The protein-free supernatant was then purified using the Maxwell® 16 MDx instrument and LEV simplyRNA Purification Kit (Promega, Madison, WI), according to the manufacturer specifications. The transcriptome library preparation and sequencing were performed by Novogene (HK) Company Limited. RNA degradation and contamination was monitored on 1% agarose gels, RNA purity was checked using the NanoPhotometer® spectrophotometer (IMPLEN GmbH, Munchen, Germany) and RNA concentration was determined using the Qubit® RNA Assay Kit in Qubit® 2.0 Fluorometer (Life Technologies, Carlsbad, CA). RNA integrity was assessed using the RNA Nano 6000 Assay Kit of the Agilent Bioanalyzer 2100 system (Agilent Technologies, CA, USA). A total amount of 3 μg RNA per sample was used as input material for the RNA sample preparations. Library preparation for strand-specific transcriptome sequencing was generated using NEBNext® Ultra™ Directional RNA Library Prep Kit for Illumina® (NEB, USA) following manufacturer's recommendations. Briefly, rRNA was removed using a specialised kit that leaves the mRNA, followed by the RNA fragmentation. First strand cDNA was synthesised using random hexamer primer and M-MuLV Reverse Transcriptase and second strand cDNA synthesis was subsequently performed using DNA Polymerase I and RNase H. In order to select cDNA fragments of preferentially 150–200 bp in length, the library fragments were purified with AMPure XP system (Beckman Coulter, Beverly, MA). Library quality was assessed on the Agilent Bioanalyzer 2100 system. The clustering of the index-coded samples was performed on a cBot Cluster Generation System using the HiSeq PE Cluster Kit cBot-HS (Illumina) according to the manufacturer's instructions. After cluster generation, the library preparations were sequenced on a Hiseq 400 (Illumina) and paired-end reads were generated. Raw reads were mapped to the genome with Bowtie2 2.3.4.1[47]. The files were further converted with samtools 1.6[48], and read counts were obtained with htseq-count 0.6.1p1[69]. Differential expression analysis was performed with R version 3.5.3[70] and DESeq2 1.22.2[71]. Genes with a *p* value < 0.05 and log2 fold change >1 were considered relevant. EC numbers of these genes were matched with matplotlib[72] onto the maps of the KEGG database[66] for further analysis. The raw data have been uploaded to the ENA, under accession numbers ERR3262781 to ERR3262792 (BioProject PRJEB22313).

**Proteomic analysis.** For proteomics analysis, cells were cultivated under autotrophic and heterotrophic (acetate/$H_2$/$CO_2$/sulphate) conditions as described above. Each condition was performed in four biological replicates. Fifty millilitres of culture were harvested at late exponential growth phase (79% and 76% of the maximum $OD_{600}$ for autotrophic, and heterotrophic (acetate/$H_2$/$CO_2$/sulphate), respectively), centrifuged for 10 min at 10000 $g$ at 4 °C, the pellet washed with 1 ml PBS and transferred to a protein low-binding eppendorf tube. Samples were then centrifuged (10 min, 4 °C, 10000 $g$) and the cell pellets stored at −80 °C. Cell pellets were then resuspend in 0.5 ml of 100 mM Tris/HCl pH 7.5 and the suspension was sonicated ten times using a Branson sonifier SFX150 equipped with a 3-mm tip (Branson, Carouge, CH), 20% amplitude in cycles of 30 s pulse and 30 s rest on ice.

Unbroken cells and cell debris were removed by centrifugation at 10000 $g$ for 10 min at 4 °C and the protein concentration in the supernatant was measured with the Pierce BCA Protein Assay Kit (Thermo Scientific, Waltham, MA). Ten micrograms of protein were loaded on a C18 Empore stagetip, followed by a washing step with 100 μl of ammonium bicarbonate buffer (ABC) and 100 μl of 5% acetonitrile in ABC buffer. Disulphide bonds of the proteins were dissociated by incubating with 10 mM dithiothreitol in 50 mM ABC at 60 °C for 1 h. Carbox-amidomethylation of the reduced cysteines was performed with 20 mM iodoacetamide in 100 mM Tris (pH 8) at room temperature in dark conditions for 1 h. After a clean-up with 100 μl of ABC buffer, stage-tips were moved to clean 0.5 ml low-binding microcentrifuge tubes, and enzymatic digestion was performed by adding 500 ng of trypsin in 20 μl of ABC (50 mM) on top. The tryptic digestion was incubated at room temperature shaking at 50 rpm for 18 h. To stop the digestion, 10% trifluoroacetic acid in $H_2O$ was added to the samples until the pH dropped to 3. The remaining liquid was eluted through into the low-binding microcentrifuge tube followed by elution with 75 μl 0.1% formic acid in water and 5-μl 50% AcNi/ 50% 0.1% formic acid. The peptide samples were analysed after injecting (18 μl sample size) on a 0.10 × 32 mm Magic C18AQ 200A 5 μm beads pre-concentration column (Bruker Nederland) at a constant pressure of 270 bar (resulting in a flow of ~7 μL per min). Peptides were then transferred onto a 0.10 × 250 mm Magic C18AQ 200A (3 μm bead size) analytical column with an acetonitrile gradient (flow rate 0.5 μl per minute) with a Proxeon EASY nanoLC. A gradient was applied at 8–33% acetonitrile in water with 23.6 mM formic acid over 50 min, followed by a fast increase in 3 min with a percentage of acetonitrile to 80% (with 20% water and 23.6 mM formic acid in both the acetonitrile and the water) as a column cleaning step. In between pre-concentration and analytical column, a A P-777 Upchurch Microcross column was included. Within the waste line of the Microcross an electrospray (3.5 kV) was applied directly to the eluent via a stainless-steel needle. FTMS spectra were measured in full-scan positive mode between *m/z* 380 and 1400 on a LTQ-Orbitrap XL (Thermo electron, San José, CA) at high resolution (60,000). CID fragmented MSMS scans of the four most abundant 2+ and 3+ charged peaks in the FTMS scan were recorded in a data-dependent mode in the linear trap (MS/MS threshold = 5000, 45 s exclusion duration for the selected *m/z* 25 ppm). A *D. desulfuricans* protein sequence database (NCBI accession number 631220) was used together with a database of contaminants, which e.g., contains sequences of common contaminants: BSA (P02769, bovine serum albumin precursor), trypsin (P00760, bovine), trypsin (P00761, porcine), keratin K22E (P35908, human), keratin K1C9 (P35527, human), keratin K2C1 (P04264, human) and keratin K1CI (P35527, human). The label-free quantification (LFQ) as well as the match between runs options were enabled. De-amidated peptides were allowed to be used for protein quantification. Other quantification settings were kept default. Filtering and further analysis of the MaxQuant/Andromeda workflow output and the analysis of the abundances of the identified proteins were performed with the Perseus 1.5.5.3 module (available at the MaxQuant suite). Peptides and proteins were accepted for further analysis when they had a false discovery rate (FDR) of <1% and proteins with at least two identified peptides of which at least one was unique and one unmodified. Reversed hits were deleted from the MaxQuant output. The normal logarithm was taken from protein LFQ MS1 intensities as obtained from MaxQuant. Zero Log LFQ values were replaced by a value of 4.70 (a value slightly lower than the lowest measured value) to make sensible ratio calculations possible. Relative protein quantitation of autotrophic to heterotrophic (acetate/$H_2$/$CO_2$/sulphate) condition was done with Perseus by applying a two sample *T* test using the LFQ intensity columns obtained with *T*-test FDR set to 0.05 and S0 set to 1[73]. The nLC-MSMS system quality was checked with PTXQC[74] using the MaxQuant result files. The mass spectrometry proteomics data have been deposited to the ProteomeXchange Consortium via the PRIDE[75,76] partner repository with the dataset identifier PXD013114.

**Stable isotopic labelling with $^{13}$C-formate.** For the proteinogenic amino acids labelling studies, cells were grown in biological triplicates in 30 ml bottles containing 10 ml anoxic medium prepared as described above, with 30 mM sulphate, $H_2$/$CO_2$ in the gas phase, and supplemented with 75 mM sodium formate-$^{13}$C (99% $^{13}$C, Sigma-Aldrich). After reaching stationary phase ($OD_{600}$ ~ 0.18), the cells were harvested by centrifugation of 9 ml of culture for 5 min at 10,000 $g$. The cells were washed once with distilled water and then resuspended in 6 M HCl for hydrolysis at 95 °C for 24 h. Then samples were completely dried under an air stream at 95 °C. Hydrolysed samples were resuspended in 1 ml of distilled water, and centrifuged for 5 min at 10,000 $g$ to remove solid particles. Supernatants with hydrolysed amino acids were analysed with UPLC–electrospray ionisation (ESI)–MS[77]. Chromatography was performed with a Waters Acquity UPLC system (Waters, Etten-Leur, The Netherlands), using an HSS T3 C18 reversed phase column (100 × 2.1 mm, 1.8 μm; Waters, Etten-Leur, The Netherlands). The mobile phases were 0.1% formic acid in $H_2O$ (A) and 0.1% formic acid in acetonitrile (B). The flow rate was 0.4 ml min$^{-1}$ and the following gradient was used: 0–1 min 99% A; 1–5 min linear gradient from 99% A to 82% A; 5–6 min linear gradient from 82% A to 1% A; 6–8 min 1% A; 8–8.5 min linear gradient from 1% to 99% A; 8.5–11 min re-equilibrate with 99% A. Mass spectra were acquired using an Exactive mass spectrometer (Thermo Scientific, Sunnyvale, CA) in positive ionisation mode, with a scan range of 50.0–300.0 *m/z*. The spectra were recorded during the first 5 min of the LC gradients. Data analysis was performed using

Xcalibur (Thermo Scientific, Sunnyvale, CA). The identification amino acids was based on retention times and $m/z$, which were determined by analysing amino acid standards (Sigma-Aldrich, St. Louis, MI) under the same conditions.

**Dynamic isotopic labelling with $^{13}$C-formate**. For the time series metabolomics on $^{13}$C isotopic tracer experiments, cells were grown in biological triplicates in 1 L glass bottles containing 500 ml anoxic medium prepared as described above, with 30 mM sulphate and shaking at 175 rpm. The headspace was flushed daily with $H_2/CO_2$ (80/20% v/v) to avoid limitation of $H_2$ and $CO_2$. When cells were at the late exponential phase (73% of maximum $OD_{600}$), 20 ml sample was taken as time 0 and 15 mM of $^{13}$C-formate was added. After that, 20 ml samples were taken at 30 s, and 1, 2.5, 4, 5.5, 7, 9 and 24 h. The samples were immediately filtered through 0.22 μm 47 nm Nylon filters (Millipore, HNWPO4700) inserted in sterile syringe filtrating holders (Swinnex, Millipore, SX0004700). The filters were then positioned upside down in glass petri dishes, placed on dry ice, containing 2 ml of extraction solvent (40:40:20 acetonitrile/methanol/distilled water). After 5 min, the extraction solvent was transferred to 2 ml eppendorf tubes and centrifuged for 5 min at 14000 $g$ at 4 °C. The supernatant was then transferred to a clean 2 ml eppendorf tube and kept at −80 °C. Controls for measuring background $^{13}CO_2$ in the medium were taken at times 0.5, 1.5, 3, 4.5, 7.5, 9.2 and 24.5 h. For this, 3 ml of liquid samples were taken, and immediately filtered through a polyethersulfone membrane syringe filter (MDI, India) into a 50 ml anaerobic bottle containing 1 ml of 6 N HCl. Headspace gas composition was then measured on a gas chromatograph–mass spectrometer (GC-MS, DSQ MS Thermo Fisher Scientific) composed of a Trace GC Ultra (Thermo Fisher Scientific, Waltham, MA). Helium was used as a carrier gas with a flow rate of 120 ml min$^{-1}$ and a split ratio of 50. The inlet temperature was 80 °C, the column temperature was set at 40 °C and the ion source temperature at 200 °C. The fraction of $^{13}CO_2$ and $^{12}CO_2$ was derived from the mass spectrometer. Validation of the method was done using standards with known mixtures of $^{13}CO_2$ and $^{12}CO_2$. Samples were analysed using a high-performance LC (HPLC)–MS/MS system consisting of a VanquishTM UHPLC system (Thermo Scientific, US) coupled by electrospray ionisation (negative polarity) to a hybrid quadrupole high-resolution mass spectrometer (Q Exactive orbitrap, Thermo Scientific, US) operated in full-scan mode for detection of targeted compounds based on their accurate masses. Properties of full MS–SIM included a resolution of 140,000, AGC target of 1E6, maximum IT of 40 ms and scan range from 70 to 1000 $m/z$. LC separation was achieved using an ACQUITY UPLC BEH C18 (2.1 × 100 mm column, 1.7 μm particle size; part no. 186002352; serial no. 02623521115711, Waters). Solvent A was 97:3 water:methanol with 10 mM tri-butylamine adjusted to pH 8.1–8.2 with 9 mM acetic acid. Solvent B was 100% methanol. Total run time was 25 min with the following gradient: 0 min, 5% B; 2.5 min, 5% B; 5 min, 20% B; 7.5 min, 20% B; 13 min, 55% B; 15.5 min, 95% B; 18.5 min, 95% B; 19 min, 5% B; 25 min, 5% B. Flow rate was 200 μl min$^{-1}$. The autosampler and column temperatures were 4 and 25 °C, respectively. Metabolomics data were processed using Maven to obtain extracted ion chromatograms and data were corrected for natural carbon isotope abundances using AccuCor[78].

To improve separation and measurement sensitivity of specific central carbon metabolites and intracellular amino acids, samples were first derivatized with either aniline[79,80] or benzyl chloroformate[81], respectively. For aniline derivatization, samples were resuspended in 50 μl HPLC-grade water, 5 μl aniline (6 M, pH 4.5), and 5 μl N-(3-dimethylaminopropyl)-N′-ethylcarbodiimide hydrochloride (200 mg per ml). After 2 h of incubation at room temperature, 1 μl of triethylamine was added to stop the reaction. For benzyl chloroformate derivatization, samples were resuspended in 10 μl HPLC-grade water, 40 μl methanol, 5 μl of triethylamine and 1 μl benzyl chloroformate and incubated at room temperature for 30 min.

**Reporting summary**. Further information on research design is available in the Nature Research Reporting Summary linked to this article.

## Data availability
All data are available in the main text, supplementary information and data or public databases. The genome sequence of *Desulfovibrio desulfuricans* G11 and raw genome and transcriptome sequencing data are available at ENA under accession number PRJEB22313. Proteome data are available in ProteomeXchange under PXD accession number PXD013114. Source data are provided with this paper.

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

## Acknowledgements

We acknowledge Änne-Michaelis and William Newell for assistance with the LC-MS for the metabolomics experiments and Daniel Amador-Noguez for access to the LC-MS used for $^{13}$C intracellular metabolomic analysis. We thank Ines Cardoso Pereira and John van der Oost for critically reading the manuscript. This research was funded by the Netherlands Organisation for Scientific Research (NWO) through SIAM Gravitation Grant 024.002.002 and the Innovation Program Microbiology (WUR), NJC acknowledges funding from NWO through a Rubicon Grant (019.163LW.035) and a Veni Grant (VI.Veni.192.156).

## Author contributions

I.S.-A., D.Z.S. and A.J.M.S. conceived the study, designed the experiments and acquired funding. I.S.-A., A.J.M.S., N.J.C. and A.B.-E. wrote the manuscript, I.S.-A., N.J.C. and I.A.G. performed the physiological experiments in the laboratory, I.S.-A. and B.H. performed the genome and transcriptome analysis, S.B. run the LC-MS, C.E.L. and N.J.C. performed the metabolome analysis, all the authors contributed to data discussion and critically reviewed the manuscript.

## Competing interests

The authors declare no competing interests.

**Additional information**

