## [Peer Review File · Nature Communications]

REVIEWER COMMENTS

Reviewer #1 (Remarks to the Author):

The manuscript of Sanchez-Andrea describes the presence of a novel autotrophic CO₂ fixation pathway, the reductive glycine pathway, in *Desulfovibrio desulfuricans*. Although it is not clear whether this pathway can be regarded as a natural autotrophic pathway (as *D. desulfuricans* normally does not grow autotrophically and needed to be adapted to autotrophy), there is no much doubt that the organism uses this pathway for CO₂ fixation in this experimental setup. The obvious deficiency of the manuscript is the absence of any enzymatic data. However, the combination of omics data with the ¹³C-labelling experiments clearly is evident enough and suggests that the authors are correct in their proposal. The manuscript is very well written and is definitely of interest for many people in the community but also for general audience, and I fully support its publication in Nature Communications. I would like to congratulate the authors and have only a few minor suggestions to improve the manuscript.

l. 151: oxoglutarate: oxoglutarate

Up-regulation of the GS/GOGAT system could be a regulatory phenomenon. The pathway appears to be dependent on ammonia concentration and may require deregulation of ammonia transporter. This deregulation could go along with the GS/GOGAT.

oxoglutarate, alpha-ketoglutarate: please name this compound in a consistent manner.

l. 239-240: "We found that the complete rGly pathway with the serine route was present in genomes of 88 microorganisms": only the presence of genes for the putative enzymes of the pathway was shown. It cannot be said from the genome analysis, whether the function of the proteins was correctly predicted and whether they really constitute the pathway.

Reviewer #2 (Remarks to the Author):

This manuscript provides experimental evidence for autotrophic growth via the reductive glycine pathway. Autotrophic growth by this pathway had previously been proposed and even engineered but had never been demonstrated for a natural isolate. This work was performed with experimentally-evolved *Desulfovibrio desulfuricans* and follows closely on the heels of a similar discovery in *Clostridium drakei* (PNAS, March 2020, PMID: 32170009). The work combines transcriptomic and proteomic data with convincing ¹³C data for in vivo reductive glycine pathway activity. My main concerns are the treatment of the experimental evolution and aspects of manuscript presentation.

Main comments.

1. The most novel aspect of this manuscript is the discovery of autotrophic growth via a potentially native reductive glycine pathway. However, experimental evolution might have been required. A need for experimental evolution doesn't necessarily negate the novelty but the experimental evolution were downplayed and difficult to interpret. For example:

a. L57 - While there might be no obvious connection between autotrophic growth and the observed mutations it should at least be acknowledged that there is no way to be certain without moving these mutations into the ancestral strain.

b. It wasn't clear whether the evolved strain is an evolved isolate or population. Or is each replicate derived from a different evolved isolate or population? If populations were used then the % enrichment of each mutation is needed.

c. There is no discussion about the observed mutations. What are the possible effects? For example, do the tRNA mutations alter the anti-codon? Do the hypothetical proteins have any conserved domains (I can't seem to look up these proteins to perform BLAST myself).

d. How many transfers/estimated generations occurred between the ancestral strain and the evolved strain(s) used?

e. L53 – How long was the lag phase (data not shown?) Was it consistent for all lineages (consistent with slow growth, perhaps of a standing low-frequency subpopulation) or did cultures grow at different times (perhaps consistent with evolution of an enabling mutation during incubation)?

f. A non-genetic approach to gauge whether mutations are required would be to transfer cultures grown from evolved isolates under autotrophic conditions to heterotrophic conditions, and then after growth, back to autotrophic conditions. Is the long lag phase observed again? If not, then autotrophic growth likely depends on a mutation in the evolved strain.

g. L56 – 'sequenced during experiments' – does this mean after autotrophic growth occurred?

h. L314 references 'aforementioned medium' but it's not clear if this medium contained organic substrates or not.

2. However VERY recent, the discovery of a similar pathway in *C. drakei* should be acknowledged and discussed.

a. Why didn't *C. drakei* come up in the tBLASTx searches (Data set 7)?

b. The authors could consider pointing out that:

i. the reductive glycine pathway was one of two autotrophic pathways operating in *C. drakei* whereas it operated alone in *D. desulfuricans*.

ii. there are now known G+ve and G-ve representatives that use this pathway.

3. Manuscript presentation.

a. Chemolithoheterotrophic (same as chemolithoorganotrophic in the methods?) is not helpful. Combining terms like litho and hetero/organo is confusing. It is not intuitive to me as a microbial physiologist and is not well-suited for a journal that targets a broad readership. I recommend being explicit about growth conditions. For example, grown with acetate and CO₂/H₂ vs CO₂/H₂ alone (purely autotrophic conditions).

b. L46 – It is not obvious why observations of growth on formate would lead to speculation of autotrophic growth. Some organisms can assimilate formate. Clarify.

c. OD axes should be labeled something like 'Cell density (OD)', just like other axes are labeled 'description (units)'

d. Data set 1

i. I was confused as to what I was looking at. At first I thought it was the lag phase mentioned on L53 but there is no lag phase in this data. Is it the raw data for all the growth experiments? Better descriptions are needed (e.g., raw data for specific figure panels). I'm not opposed to it, but is it necessary to provide the raw data?

ii. Tab 1, Cell AE17 is empty

iii. 'Lactate Formate growth curves' tab contains no data

iv. some graphs lack axis labels; 'acumulation' not necessary and spelled incorrectly

e. Pathway analysis was described as 'pathways... were curated' (L356). Please clarify how this curation was done. Data set 2 is referred to and contains screenshots likely from Metacyc.org, but this database is not cited.

f. Dataset 2 should probably include known autotrophic pathways.

g. L74 – confusing phrasing; do the citations refer to the exceptions that fix CO₂ or the 'most' strains that do not?

- h. Fig 2, Fig S1 titles need rewording; probably move 'growth' to read 'growth conditions'
- i. L97 –clarify that genes were significantly upregulated in either transcriptomic or proteomic datasets or both for the reader to understand why some proteins that were not significant or even shown in Fig 2 were included in this discussion.
- j. GS/GOGAT is referred to as being used under ammonium-limiting conditions but this isn't necessarily true. Wouldn't GS be needed for glutamine synthesis regardless.
- k. It would be helpful to mention (~L157) that GDH usually has a low affinity for ammonium compared to GOGAT for readers that are not familiar with the conditions under which each enzyme is normally used.
- l. L101 – not all of the upregulated genes mentioned are shown in Fig 3. Please adjust appropriately. It might be helpful to save mentioning GDH, GS, and GOGAT for when they are discussed later. Otherwise I thought Fig 3 was a clean and informative figure.
- m. Fig 3 title, change to 'proposed' pathway since only expression data is presented thus far
- n. L127 – use 'suggest' rather than 'indicate' since expression data is suggestive rather than indicative of activity
- o. L157 – replace 'is' with 'is likely' since based on expression data only
- p. L151 - replace 'oxy' with 'oxo'
- q. I liked the ammonium concentration experiment to test whether a high ammonium concentration drives the reaction under C-limiting conditions. Very insightful.
- r. Fig. 4 and the associated text description were helpful.
- s. Datasets 4 and 5 have highlighting that isn't explained (e.g., red vs green?; carboxysomes highlighted)
- t. L278 –detector for sulfate?
- u. L282 – which of the RI or UV detectors was used to obtain the values shown?
- v. L289 – combine with above statement on OD to eliminate redundancy
- w. L362 – explain what threshold value was used to call a tBLASTx hit. 25% homology like on L383? Was this % ID or % similarity? Was there a minimum length criteria?

Reviewer #3 (Remarks to the Author):

MS Title: The reductive glycine pathway allows autotrophic growth of *Desulfovibrio desulfuricans*

Summary: The study demonstrates chemolithoautotrophic growth of the sulphate-reducing bacterium *Desulfovibrio desulfuricans* via the reductive glycine pathway coupled to energy conservation with hydrogen and sulphate. The reductive glycine pathway had been proposed, but had not been demonstrated in any autotroph. The functioning of the pathway is supported with genomic, transcriptomic, proteomic, metabolomic data and growth rate dependency on ammonia.

General comments: The demonstration of a naturally occurring organism growing autotrophically on a reductive glycine pathway is important, and will be of interest to a wide audience including those engaged in microbiology, metabolism, evolution and ecology. The ¹³C-labelling experiments provide the most important data, that somewhat compensates for the absence of activity measurements. A genetic analysis, perhaps on FDH, might have provided even stronger evidence, but the data presented here by the authors is overall convincing. The manuscript was well written and easy to follow. Specific points for clarification are indicated below.

The reviewer is not so concerned about suppressor mutations, but would appreciate it if the following points are clarified.

- 1) Line 54: Is it correct that the reviewer should look at the "growth curves" sheet for Data S1? Titles and/or Legends on the same sheet would have made things easier. The reviewer may have missed them.
- 2) Are the three growth curves meant to display steady growth under chemolithoautotrophic growth conditions (CO₂_1~5)? In other words, these experiments were performed after the transition from initial poor growth to stable growth (lines 52-53)?
- 3) Related to line 53, the description of inoculation (lines 269-271) should be rephrased.
- 4) Can it be presumed or has it been demonstrated that the cells grown in Fig. 1 on CO₂, after grown in the presence of acetate or with lactate, will once again display initial poor growth on CO₂?
- 5) At what point was the genome sequenced?
- 6) Has the strain with six nucleotide polymorphisms used in both growth experiments used in Figs. 1A and 1B?

Although it does not seem to be the main route here, can this organism convert serine to 2-phosphoglycerate (or 3-phosphoglycerate)? The aminotransferase homolog may not be distinguishable from other aminotransferases. Are there glycerate dehydrogenase and/or glycerate kinase homologs? This route would in any case lead to ¹³C labeling patterns similar to those obtained with serine deamination. If feasible, this could be taken up later in the discussion on distribution in nature.

Line 127: Perhaps change indicate to suggest

Line 169: delete the

Is it correct that this organism has a branched citric acid pathway terminating at 2-oxoglutarate? How is oxaloacetate formed? The meaning of the dotted arrow representing the pyruvate carboxylase reaction should be described. Does this mean that there is a pyruvate carboxylase homolog, but those for a PEP carboxykinase, PEP carboxylase (or malic enzyme) are absent?

The supplementary material should be better described.

Concerning 248897_0_supp_141897_q71d3m:

What do the X's represent in the individual columns? Is it the presence or absence of homologs? If so, what proteins were used as queries? What does pyruvate synthetase stand for, is it the pyruvate:ferredoxin oxidoreductase, or PEP synthetase/synthase?

Related to the comment above, there may be a glycerate 3-kinase. Row 1852, DSVG11_1884 in 248897_0_supp_141894_q71d3l.

Reviewer #1 (Remarks to the Author):

The manuscript of Sánchez-Andrea describes the presence of a novel autotrophic CO₂ fixation pathway, the reductive glycine pathway, in *Desulfovibrio desulfuricans*. Although it is not clear whether this pathway can be regarded as a natural autotrophic pathway (as *D. desulfuricans* normally does not grow autotrophically and needed to be adapted to autotrophy), there is no much doubt that the organism uses this pathway for CO₂ fixation in this experimental setup. The obvious deficiency of the manuscript is the absence of any enzymatic data. However, the combination of omics data with the ¹³C-labelling experiments clearly is evident enough and suggests that the authors are correct in their proposal. The manuscript is very well written and is definitely of interest for many people in the community but also for general audience, and I fully support its publication in Nature Communications. I would like to congratulate the authors and have only a few minor suggestions to improve the manuscript.

Thank you for the appreciation of the work performed.

I. 151: oxoglutarate: oxoglutarate

The typo was corrected in the text. Thank you.

Up-regulation of the GS/GOGAT system could be a regulatory phenomenon. The pathway appears to be dependent on ammonia concentration and may require deregulation of ammonia transporter. This deregulation could go along with the GS/GOGAT.

We agree with the hypothesis that the reviewer puts forward. We included this explanation now clearer in the manuscript: *“Possibly, GS and GOGAT are upregulated together with the ammonia transporter in autotrophic conditions as they are in the same operon.”*

oxoglutarate, alpha-ketoglutarate: please name this compound in a consistent manner.

The form oxoglutarate has been consistently taken.

I. 239-240: "We found that the complete rGly pathway with the serine route was present in genomes of 88 microorganisms": only the presence of genes for the putative enzymes of the pathway was shown. It cannot be said from the genome analysis, whether the function of the proteins was correctly predicted and whether they really constitute the pathway.

Indeed, we did not mean that their presence implies utilization of the pathway for autotrophic growth. To make it more explicit and avoid misunderstanding, we made some modifications in the text:

- We have included the wording “set of genes” reading now as follows: *“We found that the complete set of genes for the rGly pathway via the serine route, was present in genomes of 203 microorganisms and for the route via GR in 30”* (note that based on the suggestion of another reviewer the genome analysis was updated leading to a higher number of genomes encoding the rGly pathway enzymes).
- To further clarify that genome analysis is not a proof for the activity of these pathways, we added the following sentence: *“Our genome analysis suggests*

that these microorganisms can potentially utilize the rGly pathway for autotrophic growth, but this cannot be concluded only based on genome annotations, and requires further study.”

Reviewer #2 (Remarks to the Author):

This manuscript provides experimental evidence for autotrophic growth via the reductive glycine pathway. Autotrophic growth by this pathway had previously been proposed and even engineered but had never been demonstrated for a natural isolate. This work was performed with experimentally-evolved *Desulfovibrio desulfuricans* and follows closely on the heels of a similar discovery in *Clostridium drakei* (PNAS, March 2020, PMID: 32170009). The work combines transcriptomic and proteomic data with convincing ¹³C data for in vivo reductive glycine pathway activity. My main concerns are the treatment of the experimental evolution and aspects of manuscript presentation.

We thank the reviewer for the comments. Below we explain additional experiments that we have performed to deal with the raised concern on ‘experimental evolution’. Based on these experiments we could show that experimental evolution did not lead to the autotrophic growth phenotype.

Main comments.

1. The most novel aspect of this manuscript is the discovery of autotrophic growth via a potentially native reductive glycine pathway. However, experimental evolution might have been required. A need for experimental evolution doesn’t necessarily negate the novelty but the experimental evolution were downplayed and difficult to interpret. For example:

a. L57 - While there might be no obvious connection between autotrophic growth and the observed mutations it should at least be acknowledged that there is no way to be certain without moving these mutations into the ancestral strain.

We agree that not knowing the potential connection between the observed mutations and autotrophic growth was a shortcoming in our previous manuscript. Hence, we decided to re-sequence the genomes of both culture populations that were transferred in both autotrophic (H₂/CO₂) and heterotrophic (H₂/acetate) conditions over the 3 years of the project.

This sequencing revealed **that no unique mutations were present** in the autotrophic culture, when compared to the heterotrophic culture. All 4 mutations found in the re-sequenced autotrophic culture were also present in the culture that was transferred heterotrophically. As the mutations are present in both cultures that were independently transferred for 3 years, this indicates that these “mutations” were likely already present in the genome of the original strain. Hence, we re-analyzed the presence of these 4 SNPs in the sequence data of the initial genome that we obtained for the ancestor strain. Here, we found that 2 of the SNPs were not covered originally in the sequenced genome, due to missing Illumina reads in this region. The 2 other SNPs were present in the original genome, but not annotated as SNPs in this genome due to local low sequence quality or low SNP prevalence in the original culture.

Hence, we can conclude that the previously mentioned mutations are not connected to autotrophic growth phenotype and we have removed them in the revised

manuscript. We discuss this new and important finding briefly in the manuscript. We also have included the new sequencing data which is publicly available and refer to it in the methods section.

b. It wasn't clear whether the evolved strain is an evolved isolate or population. Or is each replicate derived from a different evolved isolate or population? If populations were used then the % enrichment of each mutation is needed.

Sequencing was performed on population level and also mutations were evaluated in % enrichment levels (all 55-99%). As our additional experimental work revealed no mutations that were unique for autotrophic conditions, the mutations (and related % enrichment) are not reported in the revised manuscript. However, to make clear that we sequenced populations, we mentioned this now in the methods section.

c. There is no discussion about the observed mutations. What are the possible effects? For example, do the tRNA mutations alter the anti-codon? Do the hypothetical proteins have any conserved domains (I can't seem to look up these proteins to perform BLAST myself).

As mentioned above none of these mutations is unique in the autotrophic culture and hence were likely already present in the ancestral strain of both culture lines. Hence, we left these mutations and their potential function out now.

d. How many transfers/estimated generations occurred between the ancestral strain and the evolved strain(s) used?

Overall, during the approximately 3 years course of the project, the cultures were transferred up to 35 times. Other data presented in the manuscript were performed earlier, e.g. proteomics was performed after 15 transfers, transcriptomics after 20 transfers, and metabolomics after 31 transfers.

However, as we now explain better in the manuscript, transfers did not lead to genetic laboratory evolution of the original strain (the genome of the strain grown in autotrophic conditions did not show any mutations when compared with the strain maintained in heterotrophic conditions).

As we prove that these cultures did not evolve, the number of transfers seems not relevant to discuss in detail in the manuscript. Therefore, we just mention that: *"no genetic mutations could be identified in the genome sequence of the autotrophic culture after 35 transfers when compared with the heterotrophic culture (both after 3 years of subcultivation)"*.

e. L53 – How long was the lag phase (data not shown?) Was it consistent for all lineages (consistent with slow growth, perhaps of a standing low-frequency subpopulation) or did cultures grow at different times (perhaps consistent with evolution of an enabling mutation during incubation)?

Initially the culture was inoculated from a heterotrophic culture with 1% inoculum leading to a lag phase of 7-10 days. It was observed that when the cultures were inoculated with 10% inoculum from an autotrophic preculture in exponential phase, the lag phases were consistently short (up to 2 days). So, to prevent long lag phases, before each experiment, 10% fresh inoculum was used taken at approximately 70% of the maximum optical density. We now mention this clearly in the main text: *"Periodical transfer of the autotrophic cultures in the late exponential cultures (with 10% of inoculum) successfully avoided long lag phases."*

f. A non-genetic approach to gauge whether mutations are required would be to transfer cultures grown from evolved isolates under autotrophic conditions to heterotrophic conditions, and then after growth, back to autotrophic conditions. Is the long lag phase observed again? If not, then autotrophic growth likely depends on a mutation in the evolved strain.

That is a very nice suggestion. We started a slightly different experiment in a similar direction. Due to the corona lockdown, we were just able to properly finalize this experiment. In this experiment, we took both independent culture lines that had been transferred over the course of 3 years either in autotrophic or heterotrophic conditions. We transferred both cultures two consecutive times in autotrophic conditions and followed/compared their growth. In this way, the heterotrophic line was transferred to autotrophic conditions allowing one quick adaptation culture and we followed growth during the second adaptation. This mimicked the initial transfer but this time the transfer was done at 70% exponential phase and with 10% inoculum, whereas in the beginning of the 3 year project we used 1% inoculum and inoculum in latent phase). We observed for both a similar short lag phase (~1-2 days) during the second transfer. The fact that the 3-years adapted autotrophic line (which could have evolved) behaves the same as the heterotrophic line just transferred twice to autotrophic conditions, further confirms that no genetic adaptation of the strains happened during ~35 autotrophic transfers. The initial long lag phase we observed at the start of the project is then related to the heterotrophic preculture, its growth phase and inoculum size, and shows that *Desulfovibrio desulfuricans* G11 needs some adaptation time when transferring from heterotrophic to autotrophic conditions (likely to upregulate autotrophic pathways).

We briefly mention this experiment now and provide the related data to the second transfer in Data S1 and below here: “Furthermore, the heterotrophic population was transferred to autotrophic conditions and after two transfers showed similar growth to the long-term transferred population (Data S1).”

g. L56 – ‘sequenced during experiments’ – does this mean after autotrophic growth occurred?

At the beginning of the experimental work, we sequenced the ancestral G11 strain in our lab (before autotrophic cultivation) with PacBio (3/4/2017) and corrected with

Illumina. The other sequence data included in the revised manuscript are of both culture lines grown heterotrophically and autotrophically after 3 years of transfers. We now explain this throughout the manuscript and in the methods.

h. L314 references 'aforementioned medium' but it's not clear if this medium contained organic substrates or not.

This has been specified now to avoid confusion: "For DNA extraction, 50 ml of cell culture grown in the aforementioned medium in heterotrophic conditions (acetate/ H₂) was collected by centrifugation for 5 min at 15000 g."

2. However VERY recent, the discovery of a similar pathway in *C. drakei* should be acknowledged and discussed.

Thank you for pointing out this paper. It was published while this manuscript was in the review process, so it was not included in the discussion yet. We added a short discussion and reference to this work.

a. Why didn't *C. drakei* come up in the tBLASTx searches (Data set 7)?

The genome search used the database from 2017. *C. drakei*'s genome was published only in 2018, therefore it was missing in our search. We have now downloaded the latest version of the database (2020) and repeated the search. The new findings have been updated in the manuscript, and does now indeed include *C. drakei*, and also several other additional microorganisms.

b. The authors could consider pointing out that:

i. the reductive glycine pathway was one of two autotrophic pathways operating in *C. drakei* whereas it operated alone in *D. desulfuricans*.

This has been included in the discussion in "Recently, it was observed that the acetogen *C. drakei* combines the glycine reductase route of the rGly pathway, with the reductive acetyl-CoA pathway during autotrophic growth on H₂/CO₂, in the presence of yeast extract³⁴."

ii. there are now known G+ve and G-ve representatives that use this pathway.

According to the reviewer's suggestion, we added the following sentence: "In summary, our and other recent findings suggest that the rGly pathway is phylogenetically widespread allowing autotrophic growth in Gram-negative bacteria as shown here for *D. desulfuricans*, and in Gram-positive bacteria, such as anaerobic *Clostridia* (in cooperation with the reductive acetyl-CoA pathway)^{30,34}"

3. Manuscript presentation.

a. Chemolithoheterotrophic (same as chemolithoorganotrophic in the methods?) is not helpful. Combining terms like litho and hetero/organo is confusing. It is not intuitive to me as a microbial physiologist and is not well-suited for a journal that targets a broad readership. I recommend being explicit about growth conditions. For example, grown with acetate and CO₂/H₂ vs CO₂/H₂ alone (purely autotrophic conditions).

We agree that for a broad readership autotrophic and heterotrophic are clearer terms. In the revised main text and figures we refer to chemolithoheterotrophic growth on acetate (+H₂/CO₂) now as heterotrophic growth. Growth on H₂/CO₂ alone is generally refer as autotrophic growth. An additional control condition that we used in some

experiments was heterotrophic growth on lactate (without H₂). This is now specifically mentioned as heterotrophic growth on lactate.

In the main text we introduce this terminology now clearly: “We transferred *D. desulfuricans* to chemolithoautotrophic conditions in a basic anaerobic mineral medium¹² with sulphate and a gas phase consisting of 80% H₂ and 20% CO₂ (further referred to as **autotrophic** conditions). The inoculum was a culture growing in chemolithoheterotrophic conditions (H₂/CO₂/acetate/sulphate, further referred to as **heterotrophic**)”

b. L46 – It is not obvious why observations of growth on formate would lead to speculation of autotrophic growth. Some organisms can assimilate formate. Clarify. We understand the confusion of the reviewer. However, most pathways that support formatotrophic growth are also carbon fixation pathways supporting autotrophic growth. We now explain it more clearly in the main text.

We also included experimental data now (in Data S1) showing that indeed *D. desulfuricans* can grow purely formatotrophically (which was also confirmed by labelling to occur via the rGlyP). We mention the formatotrophic growth now specifically in the text: “We also transferred the autotrophic culture to formatotrophic conditions (formate/CO₂/sulphate) and observed good growth (Data S1).”

c. OD axes should be labeled something like ‘Cell density (OD)’, just like other axes are labeled ‘description (units)’

Thanks for the remark. We modified Fig. 1 accordingly.

d. Data set 1

i. I was confused as to what I was looking at. At first I thought it was the lag phase mentioned on L53 but there is no lag phase in this data. Is it the raw data for all the growth experiments? Better descriptions are needed (e.g., raw data for specific figure panels). I’m not opposed to it, but is it necessary to provide the raw data?

Yes, it is the raw data of the physiological experiments (Fig. 1). In answer to your comment, we now specify better the content in each tab. Raw data is recommended but not mandatory in some journals. We then leave it up to the editor if it should be included or not.

ii. Tab 1, Cell AE17 is empty

Thanks for noticing. It has been corrected since this cell corresponds to the average of two values.

iii. ‘Lactate Formate growth curves’ tab contains no data

Thanks for noticing, the lactate tab was removed as it had not relevant data for this manuscript and the missing data of formatotrophic growth is now added and referenced in the text.

iv. some graphs lack axis labels; ‘accumulation’ not necessary and spelled incorrectly

Thank you for the careful inspection of the supplementary files. The raw data graphs and axis titles have been checked (e.g. word accumulation deleted from original x-axis labels ‘time accumulation’)

e. Pathway analysis was described as 'pathways... were curated' (L356). Please clarify how this curation was done. Data set 2 is referred to and contains screenshots likely from Metacyc.org, but this database is not cited.

The screenshots are indeed from MetaCyc, which is included in the Pathway Tools software. Pathway Tools is cited in the material and methods.

f. Dataset 2 should probably include known autotrophic pathways.

Data of all known autotrophic pathways and related enzymes data are included, but we made a specific table for these, Table S3, which is also referred to in the text. We included a cross reference to this Table S3 within Data set 2 where relevant, e.g.:when describing the missing genes of TCA cycle. We think the data from Table S3 do not need to be incorporated in Dataset 2, as this dataset focusses on the biosynthesis pathways and (non-autotrophic) central metabolic pathways.

g. L74 – confusing phrasing; do the citations refer to the exceptions that fix CO₂ or the 'most' strains that do not?

The citations included references that discuss *Desulfovibrio* species that need an organic carbon source. We rephrased this more clearly:

- “Growth of *Desulfovibrio* sp. on formate and hydrogen was thought to be dependent on the presence of an organic carbon source (e.g. acetate)²⁻⁷. However, *D. desulfuricans* strain F1⁸ and the *Desulfovibrio* strains HRM1 and P23^{9,10} were reported to grow autotrophically, though insight into the CO₂ fixation pathway was not obtained.”

h. Fig 2, Fig S1 titles need rewording; probably move 'growth' to read 'growth conditions'

Thanks, the rewording recommendation has been applied to the legend.

i. L97 –clarify that genes were significantly upregulated in either transcriptomic or proteomic datasets or both for the reader to understand why some proteins that were not significant or even shown in Fig 2 were included in this discussion.

All the mentioned genes in this paragraph were upregulated in both transcriptomic and proteomic datasets: GCS, GR, FTL, ACS and GS/GOGAT; as can be seen in the figures showing proteome data (Fig. 2) and transcriptome data (Fig. 3 and Fig. S2). However, some later discussed enzymes in the proposed pathway were not always upregulated in all datasets (FHD), but this is discussed for these enzymes specifically, and their constitutive expression levels were high and may be sufficient to support the functionality of the pathway.

j. GS/GOGAT is referred to as being used under ammonium-limiting conditions but this isn't necessarily true. Wouldn't GS be needed for glutamine synthesis regardless. We refer to the combination of GS/GOGAT, indeed GS is always used for glutamine biosynthesis. But in concert with GOGAT forms an ammonia assimilation cycle, which seems upregulated in autotrophic conditions. So, we refer to the combination of GS/GOGAT consistently, as can also be seen in Figure S2. We revised and further clarified the explanation about GS/GOGAT in the revised manuscript.

k. It would be helpful to mention (~L157) that GDH usually has a low affinity for ammonium compared to GOGAT for readers that are not familiar with the conditions under which each enzyme is normally used.

We agree that the explanation would be helpful to readers and as so, it has been now included: “*Usually, ammonia assimilation GS-GOGAT, which can work with low ammonia concentrations but consumes ATP, is upregulated under ammonia-limiting conditions; whereas the GDH assimilation route is used in the presence of high amounts of ammonia and does not require ATP.*”

l. L101 – not all of the upregulated genes mentioned are shown in Fig 3. Please adjust appropriately. It might be helpful to save mentioning GDH, GS, and GOGAT for when they are discussed later. Otherwise I thought Fig 3 was a clean and informative figure.

Indeed, the nitrogen related genes are described in the separate figure (Fig. S2) to keep the Fig. 3 purely showing the CO₂ fixation route. We like to keep it this way to guide the reader through the manuscript.

m. Fig 3 title, change to ‘proposed’ pathway since only expression data is presented thus far

Changed.

n. L127 – use ‘suggest’ rather than ‘indicate’ since expression data is suggestive rather than indicative of activity

Changed.

o. L157 – replace ‘is’ with ‘is likely’ since based on expression data only

Changed.

p. L151 - replace ‘oxy’ with ‘oxo’

Changed.

q. I liked the ammonium concentration experiment to test whether a high ammonium concentration drives the reaction under C-limiting conditions. Very insightful.

Thank you for the appreciation.

r. Fig. 4 and the associated text description were helpful.

Thanks.

s. Datasets 4 and 5 have highlighting that isn’t explained (e.g., red vs green?; carboxysomes highlighted)

In Data S4 green refers to significantly upregulated (padj value) in CO₂ conditions and red significantly downregulated. In Data S5 the highlighted cells were those related with the pathway but the format has been now removed.

t. L278 –detector for sulfate?

A conductivity detector was used. This information has now been included in the manuscript.

u. L282 – which of the RI or UV detectors was used to obtain the values shown?

The RI detector was used to obtain the values. This information has now been included in the manuscript.

v. L289 – combine with above statement on OD to eliminate redundancy

As OD600 was measured using another device we prefer to keep these statements separated.

w. L362 – explain what threshold value was used to call a tBLASTx hit. 25% homology like on L383? Was this % ID or % similarity? Was there a minimum length criteria?

We are sorry, this sentence was left there by mistake, being the right place the other reference in line 383. We have now removed this sentence. Thanks for noticing.

Reviewer #3 (Remarks to the Author):

MS Title: The reductive glycine pathway allows autotrophic growth of *Desulfovibrio desulfuricans*

Summary: The study demonstrates chemolithoautotrophic growth of the sulphate-reducing bacterium *Desulfovibrio desulfuricans* via the reductive glycine pathway coupled to energy conservation with hydrogen and sulphate. The reductive glycine pathway had been proposed, but had not been demonstrated in any autotroph. The functioning of the pathway is supported with genomic, transcriptomic, proteomic, metabolomic data and growth rate dependency on ammonia.

General comments: The demonstration of a naturally occurring organism growing autotrophically on a reductive glycine pathway is important, and will be of interest to a wide audience including those engaged in microbiology, metabolism, evolution and ecology. The ¹³C-labelling experiments provide the most important data, that somewhat compensates for the absence of activity measurements. A genetic analysis, perhaps on FDH, might have provided even stronger evidence, but the data presented here by the authors is overall convincing. The manuscript was well written and easy to follow. Specific points for clarification are indicated below.

Thank you for acknowledging the work done and its importance.

The reviewer is not so concerned about suppressor mutations, but would appreciate it if the following points are clarified.

1) Line 54: Is it correct that the reviewer should look at the "growth curves" sheet for Data S1? Titles and/or Legends on the same sheet would have made things easier.

Data S1 has been carefully checked. The data sheet names has been better labelled and organized to avoid misunderstandings.

2) Are the three growth curves meant to display steady growth under chemolithoautotrophic growth conditions (CO2_1~5)? In other words, these experiments were performed after the transition from initial poor growth to stable growth (lines 52-53)?

Yes, the data was generated after the transition to a more stable growth. Since it was not mentioned in the text and it was redundant with other data already presented, the tab "growth curves" of Data S1 has now been removed.

3) Related to line 53, the description of inoculation (lines 269-271) should be rephrased.

The sentence has been rephrased to: "Periodical transfer of the autotrophic cultures in the late exponential cultures (with 10% of inoculum) successfully avoided long lag phases."

4) Can it be presumed or has it been demonstrated that the cells grown in Fig. 1 on CO₂, after grown in the presence of acetate or with lactate, will once again display initial poor growth on CO₂?

We were performing that experiment at the tie of the lockdown and could not finish it. Therefore we cannot show accurate data on this point.

5) At what point was the genome sequenced?

We answer this question already to reviewer 2 as follows: At the beginning of the experimental work, we sequenced the ancestral G11 strain in our lab (before autotrophic cultivation) with PacBio (3/4/2017) and corrected with Illumina. The other sequence data included in the revised manuscript are of both culture lines grown heterotrophically and autotrophically after 3 years of transfers. We now explain this throughout the manuscript and in the methods.

6) Has the strain with six nucleotide polymorphisms used in both growth experiments used in Figs. 1A and 1B?

As detailed above we re-sequenced the genomes of autotrophic and heterotrophic cultured over the 3 years of the project (autotrophic culture transferred 35 times) and compared it to a heterotrophically transferred strain. We found no different mutations. So, in the light of that analysis these SNPs are not relevant anymore.

Although it does not seem to be the main route here, can this organism convert serine to 2-phosphoglycerate (or 3-phosphoglycerate)? The aminotransferase homolog may not be distinguishable from other aminotransferases. Are there glycerate dehydrogenase and/or glycerate kinase homologs? This route would in any case lead to ¹³C labeling patterns similar to those obtained with serine deamination. If feasible, this could be taken up later in the discussion on distribution in nature.

This is an interesting suggestion; hence we inspected the genome and transcriptome/proteome to explore this possibility. The genome indeed encodes a potential alternative route for serine assimilation towards phospho-glycerate instead of serine deamination to pyruvate. *D. desulfuricans* G11 encodes enzymes to transaminate serine to hydroxypyruvate (which is coupled to the conversion of e.g. pyruvate to alanine), and reduction of hydroxypyruvate to glycerate and a glycerate kinase to generate phospho-glycerate. However, as the first dedicated enzyme for serine production (SHMT) from glycine is not highly upregulated, neither is one of the downstream enzymes (serine deaminase or any of the other enzymes in the alternative route to phospho-glycerate) we remain at the conclusion that the route via glycine reductase (which is highly upregulated under autotrophic conditions) is more likely.

For completeness we now mention this alternative route in the manuscript and point out potential genes for this: *“Alternatively, serine could be assimilated via conversion to phospho-glycerate, which could proceed via serine transamination to hydroxypyruvate (e.g. by pyruvate:serine transaminase DSVG11_0309 or other transaminases), subsequent reduction of hydroxypyruvate to glycerate (potentially by 2-hydroxyacid dehydrogenases DSVG11_0256 or DSVG11_0961) and finally the generation of phospho-glycerate by glycerate kinase (DSVG11_0656 or DSVG11_1884).”*

Line 127: Perhaps change indicate to suggest
Changed.

Line 169: delete the
Changed to the ¹³C-isotope form of the pathway intermediate formate.

Is it correct that this organism has a branched citric acid pathway terminating at 2-oxoglutarate? How is oxaloacetate formed? The meaning of the dotted arrow representing the pyruvate carboxylase reaction should be described. Does this mean that there is a pyruvate carboxylase homolog, but those for a PEP carboxykinase, PEP carboxylase (or malic enzyme) are absent?

The reviewer interpreted this figure correctly, *D. desulfuricans* has an incomplete TCA cycle. The dotted arrow indeed describes anaplerotic production of oxaloacetate and according to its genome annotation *D. desulfuricans* has a pyruvate carboxylase (DSVG11_2344) and no PEP carboxylase. We included this information in the caption of Figure 4.

The supplementary material should be better described.

Concerning 248897_0_supp_141897_q71d3m:

What do the X's represent in the individual columns? Is it the presence or absence of homologs? If so, what proteins were used as queries? What does pyruvate synthetase stand for, is it the pyruvate:ferredoxin oxidoreductase, or PEP synthetase/synthase?

The x indicates the presence of a homolog or pathway, we added a short caption to the supplementary Data S7 to clarify this.

The protein uses a reference/query indicated in the upper row, which is now clarified with a short caption. Pyruvate synthetase is indeed used as a synonym for pyruvate:ferredoxin oxidoreductase.

Details on the protein queries used are also described in the methods section.

Related to the comment above, there may be a glycerate 3-kinase. Row 1852, DSVG11_1884 in 248897_0_supp_141894_q71d3l.

We added this point in the discussion on alternative serine assimilation routes in the text as we discussed above

REVIEWERS' COMMENTS:

Reviewer #2 (Remarks to the Author):

I am generally satisfied with the responses to my comments. I remain enthusiastic about the findings. I perceive a few loose ends that I think are worth addressing.

1. The serial transfer conditions are still not clear. Perhaps a short section in the methods or a supplementary figure would help. For example, how many replicates/lineages were transferred? How many of these serially-transferred lineages were sequenced?

2. All figure legends should be self-explanatory, including supplemental data sets. This will greatly facilitate readability by allowing for skimming and cutting down on flipping back and forth. For example, the legends should include brief descriptions of the growth conditions. Dataset legends would be appreciated.

3. L68-71 - I had to read this sentence several times to understand it. Consider rephrasing.

4. The first reference to Dataset 1 is actually for the 4th tab in that dataset. This was unexpected and confused me for a while. Consider ordering tabs as mentioned in the text.

5. Dataset 4 highlighting was explained to the reviewers but not for future readers

6. Check manuscript for consistency. For example:
 - a. L103, 130, 329, 371, 382, 489 – once mentions H₂, CO₂, acetate; other times just H₂, acetate. Could be interpreted to mean 2 different conditions were used.

 - b. L123, 130, 331, 489 – lactate mentioned to be the sole C source but CO₂ was actually included (methods). I didn't see evidence that no CO₂ was used under heterotrophic conditions. Instead, formate accumulation in the presence of acetate + H₂ + CO₂ (Fig 1D) suggests that FDH is also active under heterotrophic conditions.

7. Chromatography methods - mM detection limit aren't that helpful. Peak height/area varies proportionately with the amount of sample injected. If the authors want to indicate detection limits perhaps they should indicate a minimum amount or list both the concentration minimum and the injection volume.

Reviewer #3 (Remarks to the Author):

The authors have sufficiently addressed the comments of this reviewer, and the function of the reductive glycine pathway in *Desulfovibrio desulfuricans* is convincingly supported with the provided evidence. There are no other comments. Some typographical points noticed while going through the manuscript are listed below.

L112 glycine

L116 glutamate

L143 can thus results might be rephrased

L201 GDH

L298 FDH

L333 were

L363 AGAGTTTGATCMTGGCTCAG

L364 The authors could define M/Y if necessary

L418 glutamine

L431 an

L447 if homology of 25% between the translation products of the nucleotide sequences was identified

Reference section Some empty lines present.

Point-by-point rebuttal to the reviewers

REVIEWERS' COMMENTS:

Reviewer #2 (Remarks to the Author):

I am generally satisfied with the responses to my comments. I remain enthusiastic about the findings. I perceive a few loose ends that I think are worth addressing.

1. The serial transfer conditions are still not clear. Perhaps a short section in the methods or a supplementary figure would help. For example, how many replicates/lineages were transferred? How many of these serially-transferred lineages were sequenced?

The reviewer is right that this could be specified in more detail. We performed the transfers in biological triplicates, but only one replicate was sequenced. We now clearly mention this in Methods sections, as well as after how many transfer genome sequencing, transcriptome sequencing, proteomics and metabolomics were performed.

2. All figure legends should be self-explanatory, including supplemental data sets. This will greatly facilitate readability by allowing for skimming and cutting down on flipping back and forth. For example, the legends should include brief descriptions of the growth conditions. Dataset legends would be appreciated.

We agree and modified the figure legends and supplementary data sets accordingly with more explanation and detail. To each Supplementary Data file we now added a legend sheet as the first sheet.

3. L68-71 - I had to read this sentence several times to understand it. Consider rephrasing.

We rewrote these sentences to further clarify this matter: "Hence, we again transferred a cell population, which was transferred heterotrophically for 3 years and not experienced autotrophic conditions before, into autotrophic conditions. During the first transfer it again showed a longer lag phase, but after a second autotrophic transfer in late exponential phase, directly a short lag phase and fast growth was observed, similar to the growth of the autotrophic culture that was transferred over 3 years (Supplementary Data 1). This emphasizes that the long-term transfer is not needed to reach the adaptation for fast autotrophic growth and this phenotype is likely is not based on genetic mutations."

4. The first reference to Dataset 1 is actually for the 4th tab in that dataset. This was unexpected and confused me for a while. Consider ordering tabs as mentioned in the text.

The first reference is to the first tab, but we agree this was unclear, so we relabelled the tab name of the 4th tab to make clear what experiment this reflects. Also the added data legends hopefully add further clarity.

5. Dataset 4 highlighting was explained to the reviewers but not for future readers

We apologize for not clarifying this in the Supplementary Data set and only in the rebuttal, we incorporated this change accordingly.

6. Check manuscript for consistency. For example:

a. L103, 130, 329, 371, 382, 489 – once mentions H₂, CO₂, acetate; other times just H₂, acetate. Could be interpreted to mean 2 different conditions were used.

We agree this was confusing, hence now consistently refer to acetate/H₂/CO₂/sulphate throughout the manuscript.

b. L123, 130, 331, 489 – lactate mentioned to be the sole C source but CO₂ was actually included (methods). I didn't see evidence that no CO₂ was used under heterotrophic conditions. Instead, formate accumulation in the presence of acetate + H₂ + CO₂ (Fig 1D) suggests that FDH is also active under heterotrophic conditions.

The reviewer brings up a good point and we corrected this accordingly by referring to lactate as the sole energy source, as in the heterotrophic growth conditions with lactate no H₂ was added. It may be that some CO₂ is fixed in these conditions with lactate, however we have not studied this in detail and do not deem this necessary for the message of

this manuscript, as we primarily focus on autotrophic growth vs heterotrophic growth on acetate.

7. Chromatography methods - mM detection limit aren't that helpful. Peak height/area varies proportionately with the amount of sample injected. If the authors want to indicate detection limits perhaps they should indicate a minimum amount or list both the concentration minimum and the injection volume.

The reviewer is right that we did not include the injection volumes in the methods. Now we added this information. to the revised manuscript.

Reviewer #3 (Remarks to the Author):

The authors have sufficiently addressed the comments of this reviewer, and the function of the reductive glycine pathway in *Desulfovibrio desulfuricans* is convincingly supported with the provided evidence. There are no other comments. Some typological points noticed while going through the manuscript are listed below.

We thank the reviewer for his second review round and careful inspection. All typos noticed we corrected accordingly.

L112 glycine

L116 glutamate

L143 can thus results might be rephrased

L201 GDH

L298 FDH

L333 were

L363 AGAGTTTGATCMTGGCTCAG

L364 The authors could define M/Y if necessary

L418 glutamine

L431 an

L447 if homology of 25% between the translation products of the nucleotide sequences was identified

Reference section Some empty lines present.